# Establishment of a fluorescent reporter of RNA-polymerase II activity to identify dormant cells

Rasmus Freter[1,2], Paola Falletta[2], Omid Omrani[1], Mahdi Rasa[1], Katharine Herbert[2], Francesco Annunziata[1], Alberto Minetti[1], Anna Krepelova[1], Lisa Adam[1], Sandra Käppel[1], Tina Rüdiger[1], Zhao-Qi Wang[1,3], Colin R. Goding [2✉] & Francesco Neri [1✉]

Dormancy, a reversible quiescent cellular state characterized by greatly reduced metabolic activity, protects from genetic damage, prolongs survival and is crucial for tissue homeostasis and cellular response to injury or transplantation. Dormant cells have been characterized in many tissues, but their identification, isolation and characterization irrespective of tissue of origin remains elusive. Here, we develop a live cell ratiometric fluorescent Optical Stem Cell Activity Reporter (OSCAR) based on the observation that phosphorylation of RNA Polymerase II (RNApII), a hallmark of active mRNA transcription elongation, is largely absent in dormant stem cells from multiple lineages. Using the small intestinal crypt as a model, OSCAR reveals in real time the dynamics of dormancy induction and cellular differentiation in vitro, and allows the identification and isolation of several populations of transcriptionally diverse OSCARhigh and OSCARlow intestinal epithelial cell states in vivo. In particular, this reporter is able to identify a dormant OSCARhigh cell population in the small intestine. OSCAR therefore provides a tool for a better understanding of dormant stem cell biology.

[1] Leibniz-Institute on Ageing, Fritz-Lipmann-Institute (FLI), Jena 07745, Germany. [2] Ludwig Institute for Cancer Research, Old Road Campus Research Building, University of Oxford, Oxford OX3 7DQ, UK. [3] Faculty of Biological Sciences, Friedrich-Schiller-University Jena, Jena 007743, Germany. ✉email: colin.goding@ludwig.ox.ac.uk; francesco.neri@leibniz-fli.de

Dormancy is cellular states characterized by a reversible exit from cellular proliferation. Cellular dormancy manifests low metabolic activity and mRNA synthesis. For example, memory T cells persist in a metabolically low state in the human body for many years, only to be re-activated when encountering their specific antigen[1,2]. Another important example of dormant cells are somatic stem cells (also often defined as quiescent stem cells) that are characterized by their dual potential for differentiation and self-renewal. Most somatic stem cells are thought to enter a state of relative dormancy, described as a reversible cell cycle exit, but poorly characterized in terms of molecular function[3,4]. Dormant stem cells have a high transplantation potential, are more resistant to cellular stresses, and have great potential to be used in regenerative medicine[5].

Somatic stem cells are commonly identified by surface markers, expression of fluorescent proteins from specific promoters or as label-retaining cells[6]. However, these methods only label specific stem cell lineages, apply only to a certain species (e.g., mouse) and can be technically challenging. Moreover, isolation of these cells by using surface markers assumes (1) prior knowledge, precluding the characterization of previously unknown stem cell populations, and (2) expression of stable markers under the analyzed conditions and/or treatments. Intestinal stem cells (ISCs) are located at the bottom of the intestinal crypts intermingled with the auxiliary Paneth cells and can be isolated by using a specific reporter mouse (Lgr5-GFP)[7]. Unlike the somatic stem cells of the most of other tissues, the Lgr5-positive cells are highly cycling cells; their activity maintains intestinal epithelium homeostasis generating all the intestinal epithelium cell lineages[7]. However, an intestinal dormant stem cell population, located just above the proliferative stem cell compartment, has been identified by using the mTert-GFP reporter mouse[8,9]. This cell population (detectable in many, but not all the intestinal crypts) is distinct from the Lgr5+ cell population, is slow cycling, resistant to injury, can give rise to Lgr5-expressing cells and may be regarded as a reserve stem cell population. Several other markers have subsequently been proposed to mark reserve stem cells (e.g., Bmi1, Hopx, Lrig1)[10–12]. Further study reported that these markers are also expressed in Lgr5+ cells[13]. However, by using transgenic mouse models (e.g., Bmi1- or Hopx-CreERT2), it is possible to isolate quiescent intestinal cells that are molecularly different from the Lgr5+ cells[14]. More recent studies, highlighting the high plasticity of the intestinal crypt cells, showed that the reserve stem pool might be composed by differentiated (or early differentiated cells) that can revert to a Lgr5+ stem state in case of need, for example, the Dll1+, Prox1+, or Krt19+ cells[15–17]. Collectively, these data suggest the intestinal stem compartment is highly plastic and composed of numerous potential dormant stem cell populations. Importantly, a reliable, specific and conserved marker of dormant somatic stem cells that permits their visualization and isolation remains elusive.

Even though it has been known for more than 40 years that dormant cells display a lower amount of total RNA[18], the molecular mechanism behind this phenomenon is poorly understood. mRNA is transcribed by RNA polymerase II (RNApII), containing a C-terminal domain (CTD) composed of 52 repeats of the heptapeptide YSPTSPS which is phosphorylated during mRNA transcription[19]. Initially, the unphosphorylated RNApII binds the promoter with phosphorylation of Serine 5 within the CTD repeat by TFIIH (CDK7 and Cyclin H) inducing promoter clearance and recruitment of the RNA capping enzyme. Subsequently, phosphorylation of Serine 2 of the CTD repeat by CDK9 and Cyclin T1 (CCNT1) is required for productive elongation[20]. Expression and activity of CDK9/CCNT1 are independent of the cell cycle, but rather linked to global metabolic activity of the cell[21]. CDK9 is an essential kinase for transcription of RNApII-dependent genes. Not only does it phosphorylate the CTD of RNApII to overcome proximal pausing of RNApII by phosphorylation of negative elongation factors DSIF and NELF[20], it also promotes recruitment of splicing and polyadenylation factors and release of the mature mRNA. Loss of CDK9 in Caenorhabditis elegans and Drosophila results in complete absence of de novo mRNA transcription[22,23]. Significantly, somatic stem cells of multiple lineages display low or absent RNApII Serine 2 phosphorylation (RNApII-pSer2), suggesting inactivity of CDK9/CCNT1 and global down-regulation of productive mRNA transcription in adult stem cells[24].

Since phosphorylation of RNApII is readily found in differentiated cells[24], the low RNApII phosphorylation observed in stem cells is highly specific to dormant cells and distinguishes them from other types of non-proliferating cells such as differentiated or senescent cells. These observations raise the possibility of a fluorescent reporter that could differentiate between cells with high or low RNApII-pSer2 kinase activity that would provide a tool to allow the identification and isolation of live dormant cells from any tissue.

Visualization of kinase activity in living cells has been a long-standing challenge. Most kinase reporters utilize FRET (Förster resonance energy transfer) technology, which is technically challenging and prone to high background signals[25]. Furthermore, the dynamic ranges obtained for signal/noise ratio are around 20–30%[26–28], which is too low for isolation of cells by fluorescence activated cell sorting (FACS) where a higher dynamic range should be achieved. Other classes of genetically encoded kinase reporters include kinase translocation reporters, in which kinase activity results in shuttling of fluorescent proteins between cytosol and nucleoplasm[29–31]. These kinase activity reporters have a higher dynamic range (up to 3-fold), but require microscopy to distinguish nuclear from cytoplasmic fluorescence and are thus not suitable for FACS analysis.

To overcome these barriers and to develop a sensitive and feasible tool, we design a genetically encoded kinase reporter by insertion of a short CDK9 kinase substrate directly into the backbone of the yellow fluorescent protein Venus[32]. Phosphorylation of this reporter within the peptide results in loss of fluorescence, which can be normalized to fluorescence of the red fluorescent protein mCherry expressed from a self-cleaving peptide in cis. This ratiometric sensor, termed OSCAR for Optical Stem Cell Activity Reporter, has a significantly improved dynamic range compared to previous FRET-based systems and can be used for FACS sorting of dormant intestinal cells and time-lapse microscopy.

## Results

**Development of a genetically encoded fluorescent kinase reporter.** Previous work has established that somatic stem cell populations exhibit reduced rates of protein translation reflecting their slow-cycling or dormant state[33,34]. Since translation depends on mRNA being produced by transcription we examined several tissues for adult stem cells exhibiting low levels of RNApII-Ser2 phosphorylation, a critical determinant of active transcription elongation (Fig. S1a). We were able to detect cells lacking detectable RNApII-Ser2 phosphorylation (RNApII-pSer2) in all tissues examined including the brain (Figs. 1a and S1b), haematopoietic stem cells (Fig. 1b) and small intestine of Lgr5-GFP mice[7] (Figs. 1c and S1c). Interestingly, we could detect heterogeneity within most stem cell compartments. For example, the LGR5-GFP+ cells at the base of the crypt were positive for RNApII-pSer2 (Fig. 1c, filled arrowheads), suggesting that they are metabolically active cells as previously reported[7]. However, there was a clear population of RNApII-pSer2-negative cells

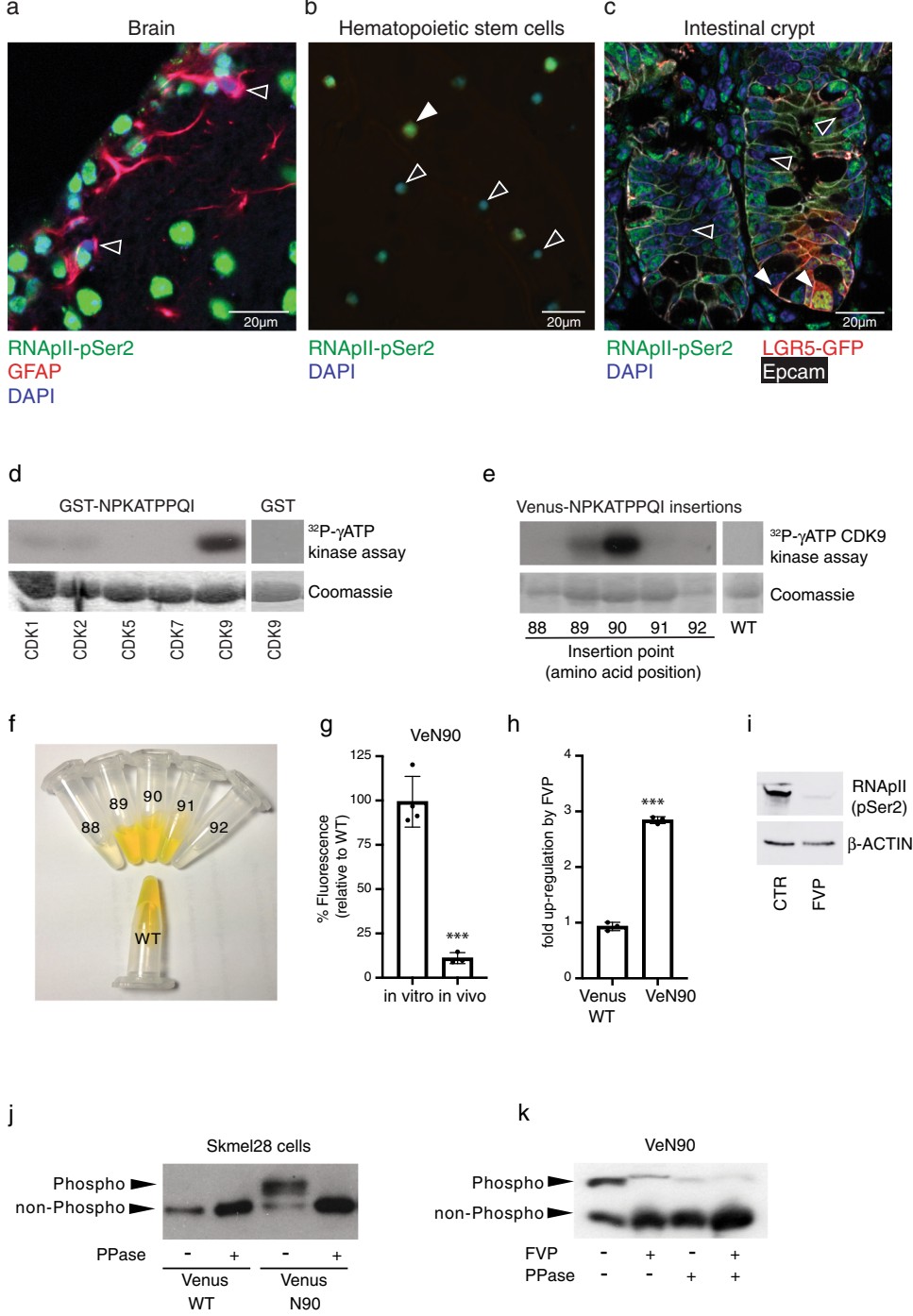

(empty arrowheads in Figs. 1c and S1c), suggesting a population of truly dormant cells above the LGR5-GFP+ cells. However, RNApII-pSer2 antibody staining neither represents a reliable approach to identify dormant stem cells (because of the possibility of false negative) nor does it allow sorting of live cells preventing their functional testing. To overcome this technical barrier, we aimed to generate a reporter for the activity of CDK9, the relevant RNApII-Ser2 kinase, by insertion of a peptide containing a CDK9-specific phosphorylation acceptor site into the backbone of a fluorescent protein. In this context, the unphosphorylated peptide insertion should not affect the fluorescence output, whereas phosphorylation of the inserted peptide should decrease or mute the fluorescence. As a consequence, high

fluorescence would positively report low CDK9 activity and therefore mark RNApII-pSer2-low dormant cells.

To select a CDK9-specific kinase target, we subjected a trypsinized and dephosphorylated HeLa whole-cell extract to an in vitro kinase assay using recombinant CDK9/CCNT1 and screened for phosphorylated peptides[35]. We obtained a preliminary list of 208 phosphorylated short peptide substrates and subsequently screened 22 redundant peptides expressed as GST fusion proteins by subjecting them to in vitro kinase assay using CDK9 and the most closely related CDKs, CDK7, CDK5, CDK2, and CDK1. Of the tested substrates only the peptide NPKATPPQI was strongly phosphorylated by CDK9, but no other CDK tested (Fig. 1d, upper panel). GST-CTD served as a positive

**Fig. 1 RNApII-pSer2 staining in dormant stem cells and characterization of the CDK9 substrate. a–c** Immunofluorescence of frozen section from mouse brain subventricular zone (**a**), FACS-purified CD150$^+$ CD34$^-$ KSL mouse hematopoietic stem cells (**b**), and frozen section from small intestine of Lgr5-eGFP mice (**c**). Antibodies were used against RNApII-pSer2, GFAP (glial fibrillary acidic protein) as a neural stem cell marker, GFP as a marker of LGR5-positive stem cells, and EpCAM, a marker of epithelial cells. In all cases, examples of RNApII-pSer2$^{low}$ cells are indicated by empty arrowheads. Filled arrowhead indicates a RNApII-pSer2$^{high}$ CD150$^+$ CD34$^-$ stem cell in hematopoietic lineage (Fig. 1b) and Lgr5+ intestinal stem cells (Fig. 1c). **d** In vitro phosphorylation of bacterially expressed and purified GST-NPKATPPQI fusion protein by indicated cyclin-dependent kinases. Empty GST protein served as negative control for CDK9 kinase. **e** In vitro kinase assay using CDK9/CCNT1 and bacterially expressed and purified Venus into which NPKATPPQI was inserted at the indicated positions. WT (wild-type) Venus served as negative control. **f** Fluorescence of bacterially expressed and purified WT Venus and derivatives with the NPKATPPQI peptide inserted at indicated positions; concentration of all proteins was 2 mg/ml. **g** Comparison of relative fluorescence of WT Venus and VeN90 expressed in 501mel human melanoma cells. Two-tailed Student's $t$-test $^{***}p < 0.0001$ ($p$ value = 0.0001), error bars indicate SEM, $n = 3$ independent batch of cell line per group were analyzed. **h** Effect of 0.5 μM Flavopiridol (FVP) on fluorescence of Ve WT and VeN90 in 501mel cells. Cells were transfected with VeN90 expression vector and 24 h later treated with 0.5 μM FVP for 48 h. Two-tailed Student's $t$-test $^{***}p < 0.0001$ ($p$ value= 4.35061E-06), error bars indicate SEM, $n = 3$ independent batch of cell line per group were analyzed. **i** Western blot using indicated antibodies of extracts from 501mel cells treated with 0.5 μM FVP for 48 h. **j** Western blot of immunoprecipitated WT Venus or VeN90 expressed in Skmel-28 cells. Immunoprecipitates were subjected to treatment with 400 units of Lambda phosphatase (PPase) before being analyzed using Phos-Tag SDS-PAGE. **k** Western blot of immunoprecipitated VeN90 expressed in 501mel cells previously treated with 0.3 μM FVP for 48 h. Where indicated, immunoprecipitates were subjected to treatment with 400 units of Lambda phosphatase (PPase) before being analyzed using Phos-Tag SDS-PAGE. All the experiments were repeated at least three times with similar results. Source data are provided as a Source Data file.

control for kinase activity against a common substrate (Fig. S1d). Having identified a CDK9 substrate peptide, we then inserted the 9 aa NPKATPPQI peptide in-frame into various positions of *Venus*[32] or *mCherry* (Fig. S1e, f). 126 different insertions were expressed and purified from bacteria and subjected to in vitro phosphorylation using CDK9/CCNT1. Most insertion sites were chosen to be within the loops of the beta-barrel structure of the fluorescent proteins since we speculated that insertion into the beta strands would abolish fluorescence completely. Since CDK9 is a nuclear kinase, a C-terminal nuclear localization sequence was added to all constructs[36] (Figs. S1e and S2a). The ability of CDK9 to phosphorylate the NPKATPPQI peptide depended strongly on the integration site. Insertion of NPKATPPQI at position 90 of Venus (VeN90) proved to be a good kinase target of CDK9, while insertion at position 91 prevented its phosphorylation (Fig. 1e). Importantly, wild-type (WT) Venus was not phosphorylated by CDK9. Of note, we could not find any insertion site of NPKATPPQI into mCherry that could be used as a CDK9 kinase sensor (Fig. S1f). We next examined whether the peptide insertion would affect Venus fluorescence. Insertion at positions 89, 90, and 91 (within the loop regions) yielded equivalent fluorescence to bacterially expressed and purified WT Venus (Fig. 1f). However, insertion at the border between the loop and the beta strands (positions 88 and 92) severely diminished basal fluorescence. Even though VeN90 purified from bacteria showed the same fluorescence as WT Venus in vitro, overexpression of VeN90 in cultured mammalian cells resulted in a strong loss of fluorescence to around 11% of that of WT Venus (Fig. 1g). To test if phosphorylation by CDK9 was responsible for this loss of fluorescence, we treated mammalian cells expressing VeN90 with Flavopiridol (FVP), a selective inhibitor of CDK9[37]. Indeed, treatment with FVP caused around a 3-fold gain in fluorescence of VeN90, but not WT Venus (Fig. 1h), accompanied by a reduction in phosphorylation of Ser2 of the RNApII CTD, the major target of CDK9 (Fig. 1i). The specificity of phosphorylation was confirmed using mass spectrometry analysis of VeN90 phosphorylated by CDK9 in vitro. The results (Fig. S2b–d) showed a single phosphorylation event at the expected Threonine within the substrate NPKATPPQI that led to a loss of basal fluorescence, including brightness as detected using fluorometric analysis (Fig. S2e–g).

To confirm that VeN90 is phosphorylated in vivo, we overexpressed FLAG-tagged Venus WT and VeN90 in mammalian cells and performed a Phos-Tag$^{TM}$ SDS-PAGE of a FLAG-immunoprecipitated cellular extract with or without prior phosphatase treatment. The Phos-Tag$^{TM}$ gel retained a phosphorylated band of the VeN90 protein that was detected neither after phosphatase treatment nor using WT Venus (Fig. 1j). These results indicate that VeN90, but not WT Venus, is phosphorylated in mammalian cells. Importantly, the upper, phosphorylated form of VeN90 was severely reduced if cells were treated with the CDK9 inhibitor FVP (Fig. 1k). Thus, inhibition of CDK9 using FVP decreases phosphorylation of VeN90 and increases its fluorescence. Nevertheless, we cannot rule out that mechanisms other than phosphorylation in vivo, such as altered protein folding, maturation of VeN90 or decreased degradation can also contribute to the change in fluorescence upon FVP treatment.

**Establishment of OSCAR, an Optical Stem Cell Activity Reporter.** An increase of CDK9 activity leads to both an increase in transcription and phosphorylation of the transgene VeN90; phosphorylation of VeN90 leads to decreased fluorescence that may be masked by a higher level of VeN90 transcript (and relative protein) due to an increased transcription. Thus, to function well as a CDK9 reporter, the fluorescence of the protein needs to be normalized to expression. We therefore generated a plasmid with a WT mCherry in-frame with triple FLAG-tagged VeN90 connected by a self-cleaving P2A peptide sequence (Fig. 2a). mCherry-P2A-VeN90 is transcribed as a single mRNA and is cleaved co-translationally, resulting in two proteins in a 1:1 stoichiometric ratio from one mRNA. In cells with active CDK9, VeN90 is phosphorylated and dim, with mCherry fluorescence being unaffected by CDK9 activity resulting in cells appearing bright red. In dormant cells with reduced or absent RNApII-pSer2 phosphorylation, a global shut-down of the majority of mRNA transcription occurs that results in low VeN90 and mCherry mRNA and protein expression. In this case, both mCherry and VeN90 mRNA and protein expression may decrease leading to a diminished fluorescence signal. However, unlike mCherry, the reduced VeN90 fluorescence arising as a consequence of lower mRNA/protein expression would be offset owing to loss of the inhibitory phosphorylation on VeN90, resulting in increased green fluorescence. As a consequence, dormant cells would have a high green/red fluorescence ratio. Activated cells, such as transit-amplifying (TA) cells, would appear yellow as both mCherry and VeN90 would be expressed with the intensity of the green fluorescence depending on the degree of VeN90 phosphorylation. As such, the ratio of green-to-red fluorescence can be used to assess CDK9 activity, and cells with high VeN90 and low mCherry fluorescence may be enriched

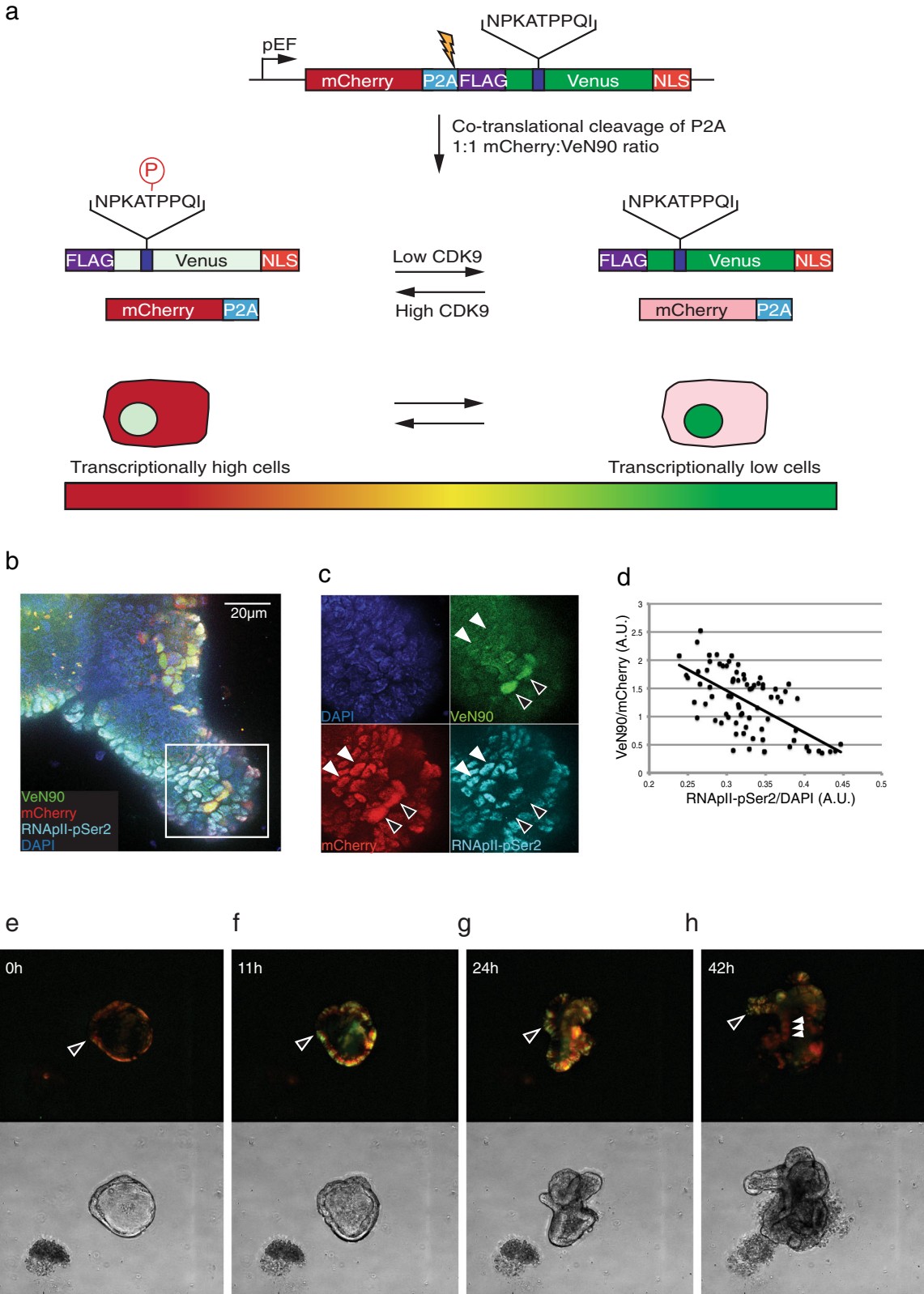

for a RNApII-pSer2 low dormant population. We termed this bicistronic ratiometric construct, consisting of *mCherry* and *VeN90*, OSCAR for Optical Stem Cell Activity Reporter. OSCAR[high] cells are defined as cells with high VeN90/mCherry fluorescence ratio, while OSCAR[low] cells as cells with low VeN90/mCherry fluorescence ratio.

To validate the ratiometric CDK9 reporter, we infected mouse small intestinal organoids with a lentivirus expressing OSCAR and detected fluorescence of VeN90, mCherry and antibody staining for RNApII-pSer2 (Fig. 2b). We observed that cells high for VeN90 fluorescence and positive for mCherry stain negative/low for RNApII-pSer2 (empty arrowheads, Fig. 2c), while cells

**Fig. 2 Characterization of OSCAR in vitro. a** Design of the OSCAR ratiometric reporter vector. Wild-type mCherry and VeN90 are expressed from a single mRNA encoding a self-cleaving P2A peptide and a nuclear localization signal fused to VeN90. During translation, mCherry and VeN90 proteins are cleaved in a 1:1 ratio allowing for normalization of VeN90 expression even if VeN90 is phosphorylated by CDK9/CCNT1 and dim in active cells. Absence of CDK9/CCNT1 in dormant cells results in gain of green fluorescence due to VeN90 lack of phosphorylation. Transcriptionally low (dormant) cells thus appear more green that non-dormant cells. Low transcription in dormant cells leads to reduced mRNA expression of the reporter leading to low mCherry and VeN90 protein expression, but reduced phosphorylation of VeN90 enhances (or rescues) its fluorescence intensity. **b** Representative image of a mouse small intestinal organoid infected with a lentivirus encoding OSCAR. After infection, the organoid was fixed, permeabilized and stained for RNApII-pSer2 and DAPI and visualized for endogenous mCherry and VeN90 fluorescence. Scale bar = 20 μM. **c** Enlarged image corresponding to white box in (**b**). Empty arrowheads indicate cells with high VeN90/Low RNApII-pSer2, while filled arrowheads indicate RNApII-pSer2-positive cells showing low VeN90 fluorescence. **d** Single cell quantification of the fluorescence signal of all fluorescent cells in (**b**) shows inverse correlation of OSCAR (VeN90/mCherry) to RNApII-pSer2 normalized to DAPI. **e–h** Time-lapse imaging of a mouse small intestinal crypt infected with OSCAR lentivirus (movie in Supplementary Information). Upper panels correspond to fluorescent images of the brightfield images below. All the experiments were repeated at least three times with similar results. Source data are provided as a Source Data file.

low for VeN90 but positive for mCherry stain high for RNApII-pSer2 (filled arrowheads, Fig. 2c). Quantification of OSCAR fluorescence ratio (defined as VeN90/mCherry fluorescence, abbreviated as OSCAR) to RNApII-pSer2 staining (normalized to DAPI to avoid out-of-focus effects) (Fig. 2d) clearly showed a negative correlation, with high OSCAR associated with low RNApII-pSer2 staining.

We next used small intestinal organoid cultures infected with OSCAR lentivirus to follow the dynamics of fluorescence activity with time-lapse microscopy. Directly after plating, all cells were OSCAR$^{low}$ (VeN90$^{low/neg}$ mCherry$^{high}$) (Figs. 2e–h and S3a, and Supplementary Movie 1). Within 11 h of culture, cell clusters at discrete spots began upregulating VeN90 fluorescence while mCherry fluorescence stayed at similar levels (empty arrowhead, Fig. 2f). New crypt-like structures formed at all of these VeN90$^{high}$ spots, as highlighted at 24 h (Fig. 2g). Interestingly, 42 h after start of microscopy, a fully developed crypt had appeared bearing three single green fluorescent OSCAR$^{high}$ (VeN90$^{high}$ mCherry$^{low}$) cells located at the base of the bud (crypt-like structure; empty arrowhead, Fig. 2h), yellow OSCAR$^{med}$ (VeN90$^{high}$ mCherry$^{high}$) cells at the middle of the bud and red fluorescent OSCAR$^{low}$ (VeN90$^{low/neg}$ mCherry$^{high}$) cells migrating away from the crypt-like structure (filled arrowheads, Figs. 2h and S3a, and Supplementary Movie 1). Importantly, expression of mCherry is not lost in OSCAR$^{high}$ cells. Overexposure of OSCAR$^{high}$ cells appearing green at the tip of the crypt shows that in fact these cells do retain a low level of mCherry fluorescence (Fig. S3b). These cells most likely downregulate mCherry and VeN90 mRNA expression as a consequence of low CDK9-dependent transcription, while VeN90 fluorescence is gained due to its lack of phosphorylation. This shows that crypts contain cells that have different ratios of VeN90 to mCherry fluorescence, presumably indicating different activity of the CDK9 kinase able to phosphorylate the VeN90 reporter.

**Development of the OSCAR mouse model.** We identified absence of RNApII-pSer2 phosphorylation as a conserved marker for dormant stem-like cells in at least seven lineages[24], suggesting that the OSCAR system would be useful for identification and isolation of dormant cells from live tissues as well. These preliminary experiments showed that this reporter may rely not only on the brightness of the VeN90 fluorescent protein (due to the phosphorylation state), but also on the general expression level of both the fluorescent proteins. Therefore, experiments involving transient or viral-mediated overexpression of the fluorescent proteins may suffer from consistent transgene expression variability among different cells. To overcome this issue, we generated the OSCAR mouse model by targeted knock-in of the *pEF1a-loxP-Stop-loxP-OSCAR* construct into the Rosa26 locus. After breeding with a Deleter-Cre(Vasa-Cre) mouse line, which has Cre

activity in germ cells[38], mice expressing OSCAR from the EF1a promoter in all tissues were derived. This mouse line is called EF1a-OSCAR (OSCAR mouse). Analysis of endogenous fluorescence of the OSCAR construct in the small intestine and co-staining of RNApII-pSer2 showed that OSCAR$^{high}$ (VeN90$^{high}$ mCherry$^{low}$) cells have low or no staining of RNApII-pSer2 (arrowheads, Fig. 3a). RNApII-pSer2 signal in OSCAR$^{low}$ cells was significantly higher than in OSCAR$^{high}$ cells (Fig. 3b). Conversely, RNApII-pSer2$^{high}$ cells showed significantly lower OSCAR signal than RNApII-pSer2$^{low}$ cells (Fig. 3c). Staining quantification reveals a strong negative correlation of RNApII-pSer2 staining to OSCAR fluorescence (Fig. 3d), thus proving that the OSCAR system can indeed label RNApII-pSer2$^{low}$ cells in vivo.

This in vivo tool allowed us to characterize the localization, cell cycle state, and transcriptional identity of cells with high (OSCAR low) or low (OSCAR high) RNApII-mediated transcriptional activity. OSCAR ratio increased from the base of the crypt (cell position 1, Fig. 3e) until reaching a maximum at cell position 8–10 (including the Paneth cells in the counting) before dropping again in the transition to the villus structure. Considering Paneth cells, this position is consistent with mTert-GFP cells (found between positions 5 and 8) and with other slowly cycling label-retaining cells distributed between positions 4 and 9, commonly referred to as "+4" cells[8]. To better characterize the cell cycle state of the OSCAR$^{high}$ cells, we performed analysis of endogenous OSCAR fluorescence with the proliferation marker Ki67 (Fig. S4a). The analysis showed that OSCAR$^{high}$ are uniformly quiescent (Ki67$^{negative}$, Fig. 3f) while Ki67$^{high}$ cells have a lower OSCAR signal compared to Ki67$^{low}$ cells (Fig. 3g).

**Isolation of the different OSCAR cell populations in the small intestinal epithelium.** FACS analysis of small intestinal crypts of EF1a-OSCAR mice showed several populations of live EpCAM$^+$ epithelial cells (for gating strategy see Fig. S4b), including OSCAR$^{high}$ (VeN90$^{high}$ mCherry$^{low}$) and OSCAR$^{low}$ (VeN90$^{low}$ mCherry$^{high}$) cells, as well as populations of cells having higher or lower general expression of OSCAR transgenes from the EF1a promoter (Fig. 4a). Populations having the same OSCAR fluorescence ratio, but different general expression of OSCAR from the Ef1a promoter were further distinguished as EF1a$^{low}$ and EF1a$^{high}$ cells. We FACS-sorted 5 different live EpCAM$^+$ populations: OSCAR$^{low}$ EF1a$^{low}$ (Population P1), OSCAR$^{high}$ EF1a$^{low}$ (P2), OSCAR$^{low}$ EF1a$^{high}$ (P3), OSCAR$^{med}$ EF1a$^{low}$ (P4), and OSCAR$^{med}$ EF1a$^{high}$ (P5) (Fig. 4b). FACS analysis of the VeN90 and mCherry intensities of the sorted populations confirmed that P2 bears high OSCAR signal, P1 and P3 low OSCAR signal and P4 and P5 medium OSCAR signal (Fig. 4c). Quantification of the cells in the different populations showed that the P4 population contains the highest number of cells (~43% of the

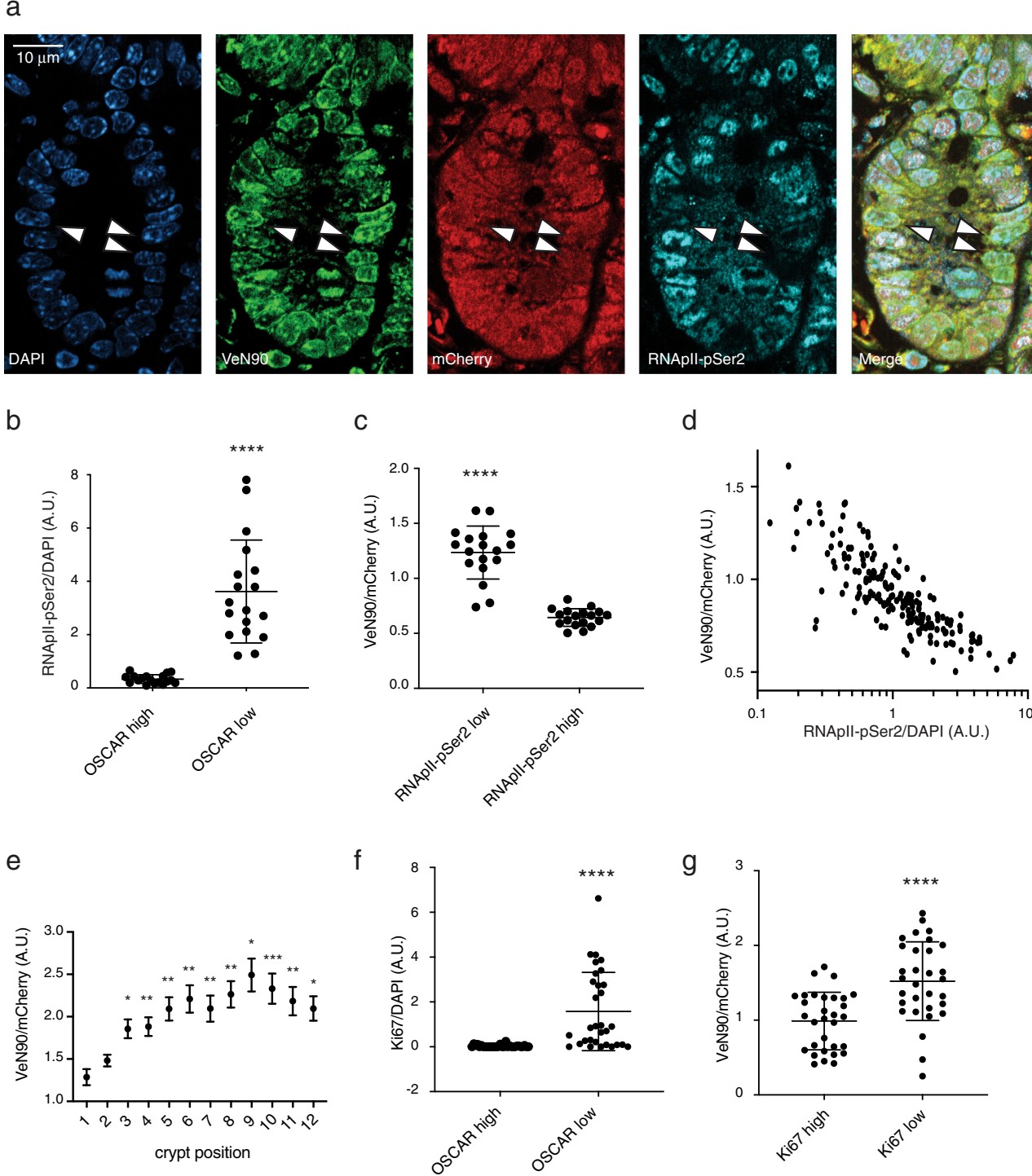

parental events) followed by the P5 (~16%), P3 (~10%) and finally P1 and P2 (both ~4%) (Fig. 4d).

We performed RNAseq analysis of the P1–P5 populations from two biological replicates. Unsupervised hierarchical clustering of the Pearson correlation between the different samples showed high correlation (Pearson coefficient >0.8) between the biological replicates (Fig. 4e). We found the P4 and P5 populations strongly clustering together ($p > 0.8$), with P2 and, even further, P1 and P3 clustering separately (Fig. 4e). To have higher resolution of the sample correlation we performed principal component analysis (PCA) that confirmed the separation of the P1 and P3 populations from the other populations and

the transcriptional proximity of the P4 and P5 populations (Fig. 4f).

**Characterization of the isolated intestinal OSCAR cell populations.** To characterize the 5 sorted OSCAR populations, we performed a geneset enrichment analysis by using different marker datasets of the major cell types in the intestinal epithelium (Fig. 5a)[39]. P4 and P5 populations showed a significant enrichment in ISC markers, while P1 population was significantly enriched in enterocyte markers (Fig. 5a). Interestingly, all the markers of fully differentiated secretory cells of the intestinal epithelium (Paneth, Goblet, Tuft, and enteroendocrine cells) were

**Fig. 3 Immunofluorescence analysis of OSCAR in vivo. a** Crypts of the small intestine of an EF1a-OSCAR mouse were stained with DAPI and anti-RNApII-pSer2 antibody. Endogenous fluorescence of VeN90 and mCherry was recorded at the same time. Arrowheads show OSCAR^high (VeN90^high mCherry^low) and RNApII-pSer2^low cells. **b** Top 10% of OSCAR^high cells stain negative for RNApII-pSer2, while the top 10% of OSCAR^low cells stain positive for RNApII-pSer2. Data are presented as mean value ± SD. ***$p < 0.0001$ ($p$ value < 0.0001). **c** Top 10% of RNApII-pSer2^low cells have significantly higher OSCAR fluorescence ratio. Data are presented as mean value ± SD. ***$p < 0.0001$ ($p$ value < 0.0001). **d** Analysis of the quantification of the OSCAR (VeN90/mCherry) and the RNApII-pSer2/DAPI in intestinal epithelium cells. Quantification in (**b**, **c**, **d**) shows data collected from 183 cells from a representative sample. Additional collected data from other 3 independent experiments and 2 mice are available in Supplementary Data 1. **e** OSCAR ratio fluorescence for each cell residing at a different position of the small intestinal crypts. The cell residing at the center of the bottom of the crypt is the number 1. The highest OSCAR has been measured at position 9. Data are presented as mean value ± SEM. *$p < 0.05$; **$p < 0.01$; ***$p < 0.001$ (Position(p) 2 vs p1, $p$ value = 0.1019, p3 vs p1, $p$ value = 0.0012, p4 vs p1, $p$ value = 0.0006, p5 vs p1, $p$ value = 0.0002, p6 vs p1, $p$ value = 0.0002, p7 vs p1, $p$ value = 0.0004, p8 vs p1, $p$ value < 0.0001, p9 vs p1, $p$ value < 0.0001, p10 vs p1, $p$ value < 0.0001, p11 vs p1, $p$ value = 0.0002, p12 vs p1, $p$ value < 0.0001. $n$ = 783 cells of 37 single crypts from 2 different mice. Statistic is calculated for each position compared to position 1. **f**, **g** OSCAR^high cells are low for Ki67 (**f**) and Ki67^high cells show lower OSCAR fluorescence ratio (**g**). Data are presented as mean value ± SD. ***$p < 0.001$ (for (**f**) and (**g**) $p$ value < 0.0001). $n$ = 31 cells (top10%) shown from a total of 315 cells analyzed from 2 different mice. Unpaired two-tailed $t$-test was used for all the analysis in this figure. Source data are provided as a Source Data file.

found strongly enriched in the P3 population indicating that this population contains an heterogenous pool of cells of the same lineage, presumably having the same transcriptional and metabolic state (Fig. 5a). In agreement with the in vitro experiment in organoids (Fig. 2e–h), the OSCAR^low cells (P1 and P3) marked differentiated cells (enterocytes and secretory cells), while OSCAR^medium cells represented active stem cells. Plotting the gene expression levels and independent RT-qPCR of some of the most characterized markers of the different intestinal epithelium cell types confirmed the geneset enrichment analysis (Figs. 5b and S4c). Remarkably, the P2 population did not show any significant enrichment of these known markers. With these analyses we were not able to differentiate the P4 from the P5 population, however, the P5 cells showed a minor enrichment in ISC markers and a marginally higher enrichment in enterocyte markers with respect to the P4 population (Fig. 5a, top-left and top-middle panels). FACS analysis revealed that the P5 cells have a significantly higher forward and side scatter values than P4 cells (Fig. 5c, d), indicating a bigger size and granularity (the latter may indicate a higher nucleus-to-cytoplasm complexity). We then hypothesized that the difference between P4 and P5 could be linked to their cell cycle state. First, we performed another geneset enrichment analysis with markers more specific for TA cells and we found that P5 cells are enriched in these markers with respect to the P4 population (Fig. 5e). Secondly, we made a cell-cycle FACS analysis that showed the P4 cells being enriched for G1/S phase, while P5 cells are in S/G2M phase (Fig. 5f, g). These two analyses indicated that the P4 gate contains stem cells that are entering the cell cycle, while the P5 population are mainly cells that are dividing. Taken together, these analyses indicated that P1, P3, P4, and P5 cell populations are enriched with intestinal cells of specific cell lineages or cell-cycle stage, while it was not possible to specifically identify cells within the P2 gate.

**Analysis of the OSCAR^high cells in the intestinal epithelium.** To confirm that P2 cells had low transcriptional activity as indicated by high OSCAR fluorescence ratio (Fig. 4c), we measured the total amount of RNA from the sorted cell populations normalized on the number of cells (Fig. 5h). Cells in the P2 population showed the lowest cellular RNA content among all the other populations, characteristic of cells in $G_0$ (Fig. 5h)[40] and in accordance with low transcriptional activity and dormancy. To further characterize the P2 cell population, we identified differentially expressed genes in P2 with respect to all the other populations (494 genes, $p < 0.01$). Gene Ontology (GO) analysis of this geneset revealed that P2 population is enriched in signaling pathways related to RNApII-associated transcription process, bacteria/drug defence-response and development/differentiation-

associated signaling (Fig. 6a). An expression heatmap of the genes associated with the transcription process show a strong down-regulation in the P2 population, whereas part of them are upregulated in the P1/P3 and partially upregulated in P4/P5 (Fig. 6b). This analysis indicates that OSCAR system can reliably mark the cells depending on their transcriptional activity state.

Several markers were proposed to label the intestinal reserve stem cell pool (see introduction), a population of cells with stem properties (e.g., ability to regenerate intestinal epithelium in case of damage of the Lgr5+ cells) and supposed to be in a dormant state. We therefore checked the expression of these markers in our isolated OSCAR populations. We found that Tert and Krt19 marker genes are enriched in the P2 population (Figs. 6c and S5a), whereas other genes like Hopx, Dll1, Bmi1, Prox1, and Lrig1 are more enriched in the P3 population (Fig. S5b). As previously reported some of these markers are also expressed in the P4 and P5 populations that contains Lgr5+ proliferative stem cells[13]. Interestingly, Hopx, Dll1, Bmi1, Prox1, and Lrig1 are either already known markers of cells of the secretory lineage[15,16] or transcriptionally enriched in secretory precursor or differentiated cells as shown by single-cell RNAseq analysis (Fig. S5c; Hopx and Prox1 in enteroendocrine cells, Dll1 in Goblet cells, Bmi1 and Lrig1 in secretory precursor cells)[39]. Geneset enrichment analysis confirmed that the P3 OSCAR population does indeed also contain secretory precursor cells[41] (Fig. S5d). To better characterize the P2 population, we performed an upstream regulator analysis on IPA software by using the genes found upregulated in the P2 population (Fig. 6d). This predictive analysis showed several potential master regulators of the transcriptional profile of the P2 population, and two of them, specifically Hnf4a and Erbb3 (Fig. 6d) were enriched in the P2 population (Fig. 6e). Remarkably, Hnf4a has been previously shown to be a key transcription factor for enterocyte differentiation (indeed is more expressed in the P1 population, Fig. 6e) and Erbb3 marks non-proliferative cells localized in the upper part of the intestinal crypt[42,43]. Collectively these data suggest that the P2 population, marked by Tert, Krt19 and Erbb2/3, are probably closer to the enterocyte lineage and most importantly are dormant (not cycling and transcriptionally inactive) and may work as intestinal reserve stem cell pool. To verify this hypothesis, we performed an in vitro regeneration assay by plating single sorted cells from the different OSCAR populations and tested their ability to form intestinal organoids (Fig. 6f). As expected, the P4 and P5 OSCAR populations (active stem cells) showed the highest cyst-forming efficiency at day 5 (respectively, 1.41 ± 0.70 and 1.79 ± 1.10, in percentage, average ± standard deviation), while the P1 population showed no or very low efficiency (0.02 ± 0.03) (Fig. 6f). The P2 population had a significantly higher

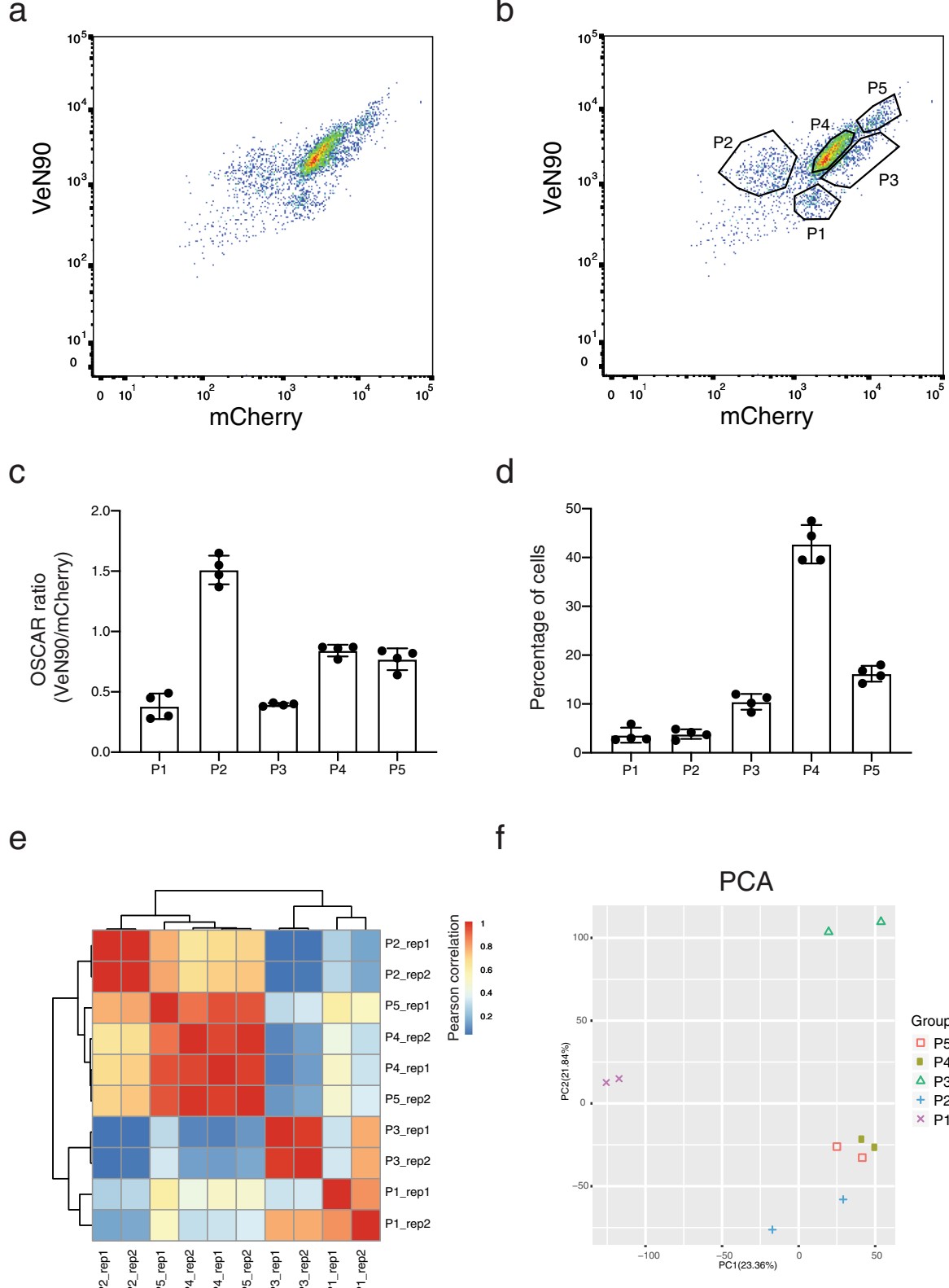

cyst-forming capacity than P1 (0.15 ± 0.08, Fig. 6f) suggesting that indeed in this population there are cells able to grow and potentially (re)generate intestinal epithelium. Interestingly, the cysts derived from P2 showed a different shape and a smaller size at day 5 when compared to the P4/P5 cysts (Fig. S6), suggesting that these cells (1) are not a contamination with P4/P5 cells and

(2) probably require more time to re-enter in the cell cycle. A reserve stem cell population has been shown to be resistant to DNA damage mediated by irradiation[8,44]. Therefore, we performed total body gamma irradiation on OSCAR mice and measured the apoptosis marker (Annexin V) by FACS analysis (Fig. 6g). Actively cycling cells (P4/5 populations) showed higher

**Fig. 4 FACS analysis of OSCAR in vivo. a** FACS analysis of EF1-OSCAR mouse small intestinal EpCAM+ live cells shows several populations of OSCAR[high] (VeN90[high] mCherry[low]) and OSCAR[low] (VeN90[low] mCherry[high]). In addition, populations with a higher or lower general transgene expression can be detected. **b** FACS gating strategy used for FACS sorting of populations P1 to P5. **c** Population P2 contains OSCAR[high] cells. Populations P1 and P3 contain OSCAR[low] cells, with P3 having higher VeN90 and mCherry fluorescence. Populations P4 and P5 are OSCAR[med], with P5 having higher VeN90 and mCherry fluorescence. $n = 4$ mice were analyzed. Data are presented as mean value ± SD. **d** Percentage of cells in each of the indicated gates with respect to their parental gate. $n = 4$ mice were analyzed. Data are presented as mean value ± SD. **e** Hierarchical clustering and heatmap of the Pearson correlation of the RNAseq datasets from the different OSCAR cell populations. **f** Principal component analysis (PCA) of the RNAseq datasets as in (**e**). Source data are provided as a Source Data file.

percentage of Annexin V+ cells with respect to the P2 cell population (Fig. 6h) indicating that the OSCAR[high] cells are indeed more resistant to DNA damage.

Taken together, these data indicate that the OSCAR reporter system can reliably mark cells depending on their transcriptional state and allow isolation of living cells for further characterization. Remarkably, our data indicate that the cell transcriptional state is a primary feature that can define cell identity and proliferation, differentiation capacity and lineage specification. We have tested the OSCAR reporter on the intestinal epithelium in the mouse, but it is worth speculating that it may lead to novel biological findings in other organs as well as in pathological conditions, in cancer, for example, where dormancy is an important clinical barrier to successful therapies. Additionally, since absence of RNApII-pSer2 is found also in dormant cells of other animals, including simpler organisms as *C. elegans* and *D. Melanogaster*, this tool could be useful in understanding cellular dormancy in other animals as well.

## Discussion
Somatic stem cells have a great potential for regenerative medicine and can contribute to cancer initiation and relapse. Both physiological tissue resident stem cells and cancer stem cells may lie dormant for many years. Hematopoietic stem cells in the bone marrow, for example, are only required during hematopoietic stress, such as during injury, enter the cell cycle only five times per lifetime in the mouse and between times exist in a dormant state[5]. Similarly in cancer, relapse many years after an apparently successful therapy is believed to arise from activation of long-term dormant cancer stem cells[45]. Dormant cells exhibit a reduction in global transcription owing to greatly reduced phosphorylation of Ser2 of the RNApII CTD, a modification essential for transcription elongation of most genes[4,46]. Low levels of transcription elongation in dormant stem cells makes sense, since stem cells are known to exhibit low rates of protein synthesis[33,34].

Not all stem cells are dormant and exhibit low levels of RNApII-pSer2. For example, many studies have determined that the LGR5-positive stem cell population located at the bottom of the intestinal crypt are active and proliferative stem cells[7,47]. However, our results show that they are clearly positive for the RNApII-pSer2 mark and are therefore unlikely to be dormant. By contrast, we also detected a population of cells within the crypt that are negative for RNApII-pSer2 and are therefore likely to be dormant. Although these may represent a reserve stem cell population, whether all RNApII-pSer2-negative cells are stem cells remains to be determined. Several studies reported the occurrence, in the intestinal epithelium, of a reserve stem pool presumably not proliferating or slow cycling that can replace the Lgr5+ stem cells in case of injury. Many markers have been proposed to mark the reserve stem cells[8–12]; however, these markers are either very rare or not uniquely expressed[8,13]; additionally, these cell populations have not been well characterized either in terms of whether they are in a deep dormant state or whether they are transcriptionally active. Single-cell RNAseq of the small intestine has been recently reported[39], which in theory could detect all cells, presumably also dormant cells, but it failed to

identify a clear separation of the dormant stem cell population in this tissue. It is tempting to speculate that due to stringent quality thresholds, dormant cells with a lower amount of mRNA due to absence of productive mRNA transcription are most likely not passing the threshold and will not be detected. Thus, the identification of a presumptive dormant cell population highlights a key challenge; the technical breakthrough that would allow the detection and isolation of dormant resident stem cells or cancer stem cells in a live cell population is highly desirable.

To meet this challenge, we used our knowledge that dormant cells exhibit low levels of RNApII-pSer2 to design a fluorescent reporter of CDK9 activity, the kinase phosphorylating RNApII-Ser2. The resulting reporter, OSCAR, provides a ratiometric green-to-red sensor for dormant cells and relies on the ability of CDK9 to phosphorylate a short peptide inserted into the backbone of Venus (VeN90) with phosphorylation leading to a reduction in Venus fluorescence. As a consequence, dormant cells exhibiting low CDK9 activity, thereby low RNApII-pSer2 can be visualized by increased Venus fluorescence compared to the internal mCherry control. Our data shows that the OSCAR reporter system is able to reliably mark cells with different transcriptional state and allow identification and isolation of dormant cells specifically. By using OSCAR, it was possible to visualize in intestinal organoids the dynamics of the generation of RNApII-pSer2-negative dormant cells in real time. Our data also indicate that the EF1a-OSCAR system can distinguish cells with different transcriptional identity, cell cycle, and lineage commitment.

Since, unlike LGR5-GFP, the OSCAR reporter does not rely on any cell type-specific promoter or activity, but instead reports low CDK9 activity it is likely to be generically useful for imaging and isolation of dormant cells in many tissue or cancer types. We also note that this strategy for development of a fluorescent kinase sensor by insertion of a small specific substrate directly into the backbone of a fluorescent protein can potentially be employed to derive additional sensors for other kinases.

Finally, the VeN90 reporter developed here for CDK9 activity and present in the OSCAR reporter is fundamentally different from a previously described CDK9 reporter[48]. The Fujinaga et al. CDK9 reporter is based on bimolecular fluorescence complementation, only functions in CDK9-positive cells, and is rather toxic to expressing cells. By contrast, in our case we were able to convert a negative observation (absence of RNApII-pSer2) into a positive event (green fluorescence) and could not detect any toxic effects of OSCAR expression in mammalian cells and mice so far. We anticipate therefore that OSCAR may prove a useful tool in characterization of dormant cells both in vitro and in vivo.

## Methods
Materials and oligonucleotides used in this study are provided in Supplementary Data 2.

**Microscopy.** Tissues of WT mice were fixed in cold 4% PFA overnight, washed with PBS and de-hydrated in 20% Sucrose at 4 °C overnight. After embedding in O. C.T. compound (VWR) and snap freezing in liquid nitrogen, samples were cut at 12 μm on a Leica CM3050S cryostat, air-dried, blocked with 2% skimmed milk in PBS containing 0.1% Triton X-100 and incubated with the indicated antibodies at

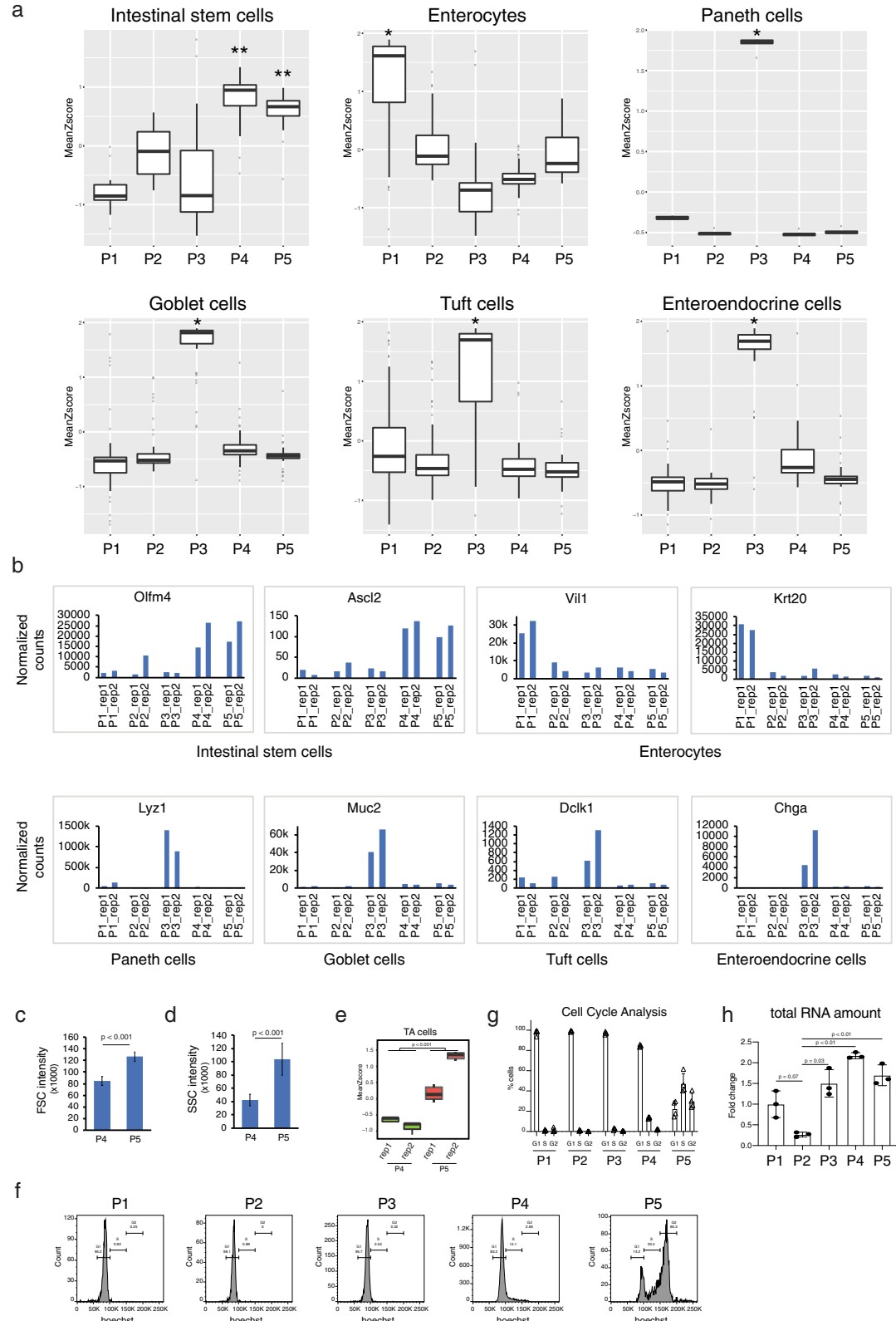

4 °C overnight. For RNApII-pSer2 staining, small intestines of EF1a-OSCAR mice were fixed in 4% PFA for 20 min at room temperature (RT) and then treated as above. Additionally, for Ki67 staining, small intestines of EF1a-OSCAR mice were fixed in 2% PFA at 4 °C overnight, and washed and de-hydrated as above; after embedding in O.C.T. compound (VWR) and snap freezing in liquid nitrogen, samples were cut at 12 μm on a Leica CM3050S cryostat, air-dried, permeabilized with PBS containing 0.1% Triton X-100 for 30 min at RT, followed by blocking with BSA 1%, 5% donkey serum in PBS containing 0.05% Triton X-100 and

incubated with Ki67 antibody in blocking buffer for 1 h at RT. Of note, endogenous OSCAR fluorescence is very sensitive to fixation, thus for other organs should be determined empirically. Antibodies used were rabbit polyclonal anti-RNA polymerase II CTD repeat YSPTSPS (phospho S2) (Abcam ab5095), mouse monoclonal anti-GFAP antibody (Abcam ab10062), Rabbit monoclonal anti-Ki67 (SP6) (Thermo Scientific RM-9106-S) and goat polyclonal anti-GFP (Rockland 600-141-215). All antibodies were used at 1:250 dilution, besides Abcam ab5095 used at 1:500. Appropriate secondary antibodies conjugated to Alexa 488, 568, or 647

**Fig. 5 Transcriptional profiling of the OSCAR intestinal cell populations. a** Box plot of geneset enrichment analysis of the indicated gene datasets in the different OSCAR cell populations. P4/P5 cell population correlates with intestinal stem cells, P1 with enterocytes and P3 with secretory cells. Boxplots shows the quartile distribution of the data. A distance of 1.5 times the inter quartile range (Q3–Q1) is measured out below the lower quartile and a whisker is drawn up to the lower observed point from the dataset that falls within this distance. All other observed points are plotted as outliers. Intestinal stem cells $n = 24$ genes, Enterocytes $n = 108$ genes, Paneth cells $n = 14$ genes, Goblet cells $n = 91$ genes, Tuft cells $n = 103$ genes, Entroendocrine cells $n = 77$ genes. Intestinal stem cells $p$ value: P1 vs P4 = 0.0002, P2 vs P4 = 0.0001, P3 vs P4 = 0.0245, P1 vs P5 = 0.0002, P2 vs P5 = 0.0006, P3 vs P5 = 0.0352, Enterocytes $p$ value: P1 vs P2 = 2.038104e − 08, P1 vs P3 = 2.931078e − 11, P1 vs P4 = 1.590139e − 11, P1 vs P5 = 4.872108e − 09, Paneth cells $p$ value: P1 vs P2, P3, P4 and P5 = 0.0039, Goblet cells $p$ value: P1 vs P3 = 1.563194e − 12, P2 vs P3 = 9.947598e − 13, P4 vs P3 = 1.791989e − 11, P5 vs P3 = 2.700062e − 13, Tuft cells $p$ value: P1 vs P3 = 2.248624e − 06, P2 vs P3 = 3.970961e − 08, P4 vs P3 = 7.164053e − 10, P5 vs P3 = 5.493567e − 10. Entroendocrine cells $p$ value: P1 vs P3 = 6.743903e − 10, P2 vs P3 = 2.273737e − 12, P4 vs P3 = 8.708412e − 09, P5 vs P3 = 4.547474e − 12. The $p$ value is calculated by Wilcoxon two-tailed test. **b** Bar charts showing the RNAseq normalized counts levels of the indicated gene markers in the analyzed OSCAR populations. **c, d** Forward (FSC) and side (SSC) scatter values from the FACS analysis in the P4 and P5 OSCAR populations. FSC is directly proportional to the size of the cells, while SSC provides information about the internal cell complexity (granules and nucleus increase SSC). $n = 4$ mice were analyzed. Data are presented as mean value ± SD (for panel (**c**), $p$ value = 0.0003 and for panel (**d**) $p$ value = 0.0028). **e** Box plot of geneset enrichment analysis of transit-amplifying (TA) cell markers in the P4/P5 cell populations. Boxplots shows the quartile distribution of the data. A distance of 1.5 times the inter quartile range (Q3–Q1) is measured out below the lower quartile and a whisker is drawn up to the lower observed point from the dataset that falls within this distance. All other observed points are plotted as outliers. The $p$ value is calculated by Wilcoxon two-tailed test. $n = 235$ gene. **f** Cell-cycle FACS analysis performed by using Hoechst33342 staining of the different cell populations isolated from the OSCAR mouse intestinal epithelial crypts. **g** Quantification of the cell-cycle FACS analysis. $n = 4$ mice were analyzed. Data are presented as mean value ± SD. **h** Bar chart indicating the relative cellular RNA content of the different cell populations isolated from the OSCAR mouse intestinal epithelial crypts. $n = 3$ pool of 2 mice were analyzed. Data are presented as mean value ± SD (for p4, $p$ value = 0.0007 and for p5, $p$ value = 0. 0.0076). Paired two-tailed $t$-test was used for the statistical analysis. Source data are provided as a Source Data file.

(Thermo Fisher Scientific) were used at 1:500 dilution before mounting (DEPEX, VWR): Alexa Fluor 488 donkey anti-mouse IgG (H + L), (A21202), Alexa Fluor 568 donkey anti-mouse IgG (H + L) (A10037), Alexa Fluor 647 donkey anti-mouse IgG (H + L) (A31571), Alexa Fluor 488 donkey anti-rabbit IgG (H + L) (A21206), Alexa Fluor 568 donkey anti-rabbit IgG (H + L) (A10042), Alexa Fluor 647 donkey anti-rabbit IgG (H + L) (A31573), Alexa Fluor 488 donkey anti-goat IgG (H + L) (A11055). Nuclei were counterstained with DAPI (Sigma). Images were taken using a Zeiss Axiovert LSM510 or Zeiss Axio Imager and images analyzed using ImageJ and Excel.

Hematopoietic stem cells were isolated from mouse femur by flushing with cold PBS and enriched using c-Kit positive selection (1.5 µl per 100 million cells of APC-conjugated rat monoclonal anti-c-Kit antibody 2B8, eBioscience 17-1171-82, and 1.5 µl per 100 million cells of anti-APC microbeads bound to LS column, Miltenyi Biotec 130-090-855) following manufacturers' protocol. Lineage positive cells (CD3, Ly-6G/C, Cd11b, B220, TER-119) were stained using Biotin anti-mouse lineage panel (1 µl per 10 million cells each, BioLegend 133307) and Streptavidin APC-Cy7 (0.5 µl per 10 million cells, BioLegend 405208). Cells were further stained with rat monoclonal anti-CD34-FITC (clone RAM34, eBiosciences 11-0341-82), anti-c-Kit-APC (clone 2B8, eBioscience), rat monoclonal anti-Sca1-PE-Cy7 (clone D7, eBiosciences 25-5981-82), and rat monoclonal anti-CD150-PE (clone 9D1, eBiosciences 12-1501-82). Cells were sorted directly onto a cooled poly-L-Lysine coated cover glass, fixed by 4% Paraformaldehyde (Sigma) for 5 min at RT, blocked and stained as described above.

Quantification of images was performed by using ImageJ software. Individual cells were identified using DAPI staining. Endogenous or immunofluorescence of individual cells was then quantified for each channel separately. OSCAR signal was calculated by making the ratio between VeN90 mean values divided by mCherry mean values to normalize intercellular possible transgene expression differences and local technical artifacts. Immunofluorescence signal values (e.g., RNApII-pSer2) were normalized to DAPI signal values to avoid out-of-focus false-negative data. Statistical analysis was performed by using PRISM v.8 as indicated in the legend of each analysis.

### Identification of peptide candidates with mass spectrometry. Identification of short peptide targets of CDK9 was performed as described previously[35]. Briefly, HeLa whole-cell extracts were lysed, digested by protease treatment and dephosphorylated using phosphatases. After kinase reaction with CDK9/CCNT1, phosphopeptides were isolated using titanium dioxide microspheres and analyzed using liquid chromatography-coupled mass spectrometry.

### Plasmids and cloning. Plasmids for lentivirus infection of mammalian were derived from CSII-EF-MCS[49] by digest with Xho1 and Xba1 (37 °C for 1 h) and insertion of either Venus (a kind gift of A. Miyawaki[32]) or mCherry (Clontech) amplified by PCR using Accuprime Taq (Invitrogen) and primers Sal-Venus for and Venus-Xba rev. PCR conditions were 95 °C for 3 min, then 30 cycles of 95 °C for 10 s, 58 °C for 10 s, 72 °C for 30 s; a final extension at 72 °C for 3 min and hold at 4 °C. Subsequently, the annealed Ultramer® (IDT) oligonucleotide BsrG1-P2A-3F MCS-Not1 encoding a self-cleaving P2A peptide, a triple FLAG peptide and a

multiple cloning site was inserted using BsrG1/Xba1 digest resulting in the plasmid CSII-EF Venus-P2A3F-MCS and CSII-EF mCherry-P2A3F-MCS.

WT mCherry and Venus were then amplified by PCR (same PCR conditions as before) with primer BamH1-Venus forward and Venus-NLS EcoR1 reverse containing the nuclear localization signal PKKKRKVEDA and an EcoR1 site. mCherry or Venus were ligated into CSII-EF Venus-P2A3F-MCS and CSII-EF mCherry-P2A3F-MCS, respectively, in-frame with the upstream coding sequence after BamH1/EcoR1 digest, resulting in plasmids CSII-EF Venus-P2A-3F-mCherry and CSII-EF mCherry-P2A3F-Venus, which were used as WT controls.

Insertion of peptide NPKATPPQI into plasmids CSII-EF Venus-P2A3F-mCherry, CSII-EF mCherry-P2A3F-Venus, pNIC28-6His-Venus, and pNIC28-6His-mCherry was achieved by modified overlap extension PCR. Briefly, the reverse primer contained 18–20 nucleotides in reverse complement 5′ to the integration site followed by 20 nucleotides encoding amino acids NPKATP. The forward primer consisted of a 13 nucleotide overlap with the 3′ end of the reverse primer, nucleotides encoding for amino acids PQI and 18–20 nucleotides of the plasmid sequence 3′ to the integration site. PCR and Dpn1 digest were performed using QuickChange Site-Directed Mutagenesis Kit (Agilent) according to manufacturer's protocol. PCR conditions were 95 °C for 3 min, then 20 cycles of 95 °C for 30 s, 58 °C for 10 s, 72 °C for 5 min; a final extension at 72 °C for 3 min and hold at 4 °C. All plasmid constructs were validated by sequencing.

Plasmids for screening of short peptide targets were derived from pGEX-4T1 backbone by BamH1/EcoR1 digest and ligation of annealed oligonucleotides. pGEX-CTD used as a positive control for kinase assays was a kind gift of Sonja Baumli[37]. Plasmids for bacterial fluorescent protein expression and purification were derived from pNIC28-Bsa4[50] by PCR of mCherry-NLS or Venus-NLS templates with primers pNIC Venus for and rev. Ligation-independent cloning was performed as described[50] resulting in plasmids pNIC28-6His-Venus and pNIC28-6His-mCherry, which were used as WT controls.

### Protein purification and kinase assay. Bl21(DE3) bacteria cells were electro-porated (Amaxa Nucleofector) with pGEX-4T1 (for short peptide targets) or the pGEX-CTD plasmid (as a positive control) and cultured on LB-Agar plates containing Carbenicillin. For the production/expression of pNIC28-6His-Venus, pNIC28-6His-mCherry or insertions of NPKATPPQI into these fluorescent proteins, Bl21(DE3) bacteria were cultured in the presence of Kanamycin. After culture for 16 h at 37 °C, individual colonies were picked and incubated in 50 ml of Terrific Broth Medium containing 100 µg/ml Carbenicillin or 50 µg/ml Kanamycin. Once the OD600 reached 0.6–0.8, IPTG (Sigma) was added to a final concentration of 0.5 mM and the culture incubated at 23 °C shaking at 200 rpm for overnight. Bacteria were spun at 4000g for 5 min, pellets lysed by sonication and proteins purified using Glutathione beads (for pGEX plasmids) or Ni-NTA beads (for pNIC plasmids) following manufacturers' protocol (Qiagen). GST beads were washed several times with wash buffer and stored at 4 °C without elution of the GST-peptide. Ni-NTA purified His-tagged proteins were further dialyzed twice (Slide-A-Lyzer Dialysis Cassettes, 10MWCO, Thermo Fisher) against 100× the volume of 50 mM Tris-HCl pH 7.9, 300 mM NaCl, 0.1 mM DTT and concentrated with Spin-X columns (Corning). Protein concentration was determined by absorbance at 280 nm (NanoDrop), and spectral properties recorded on a SpectraMax M2 (Molecular Devices).

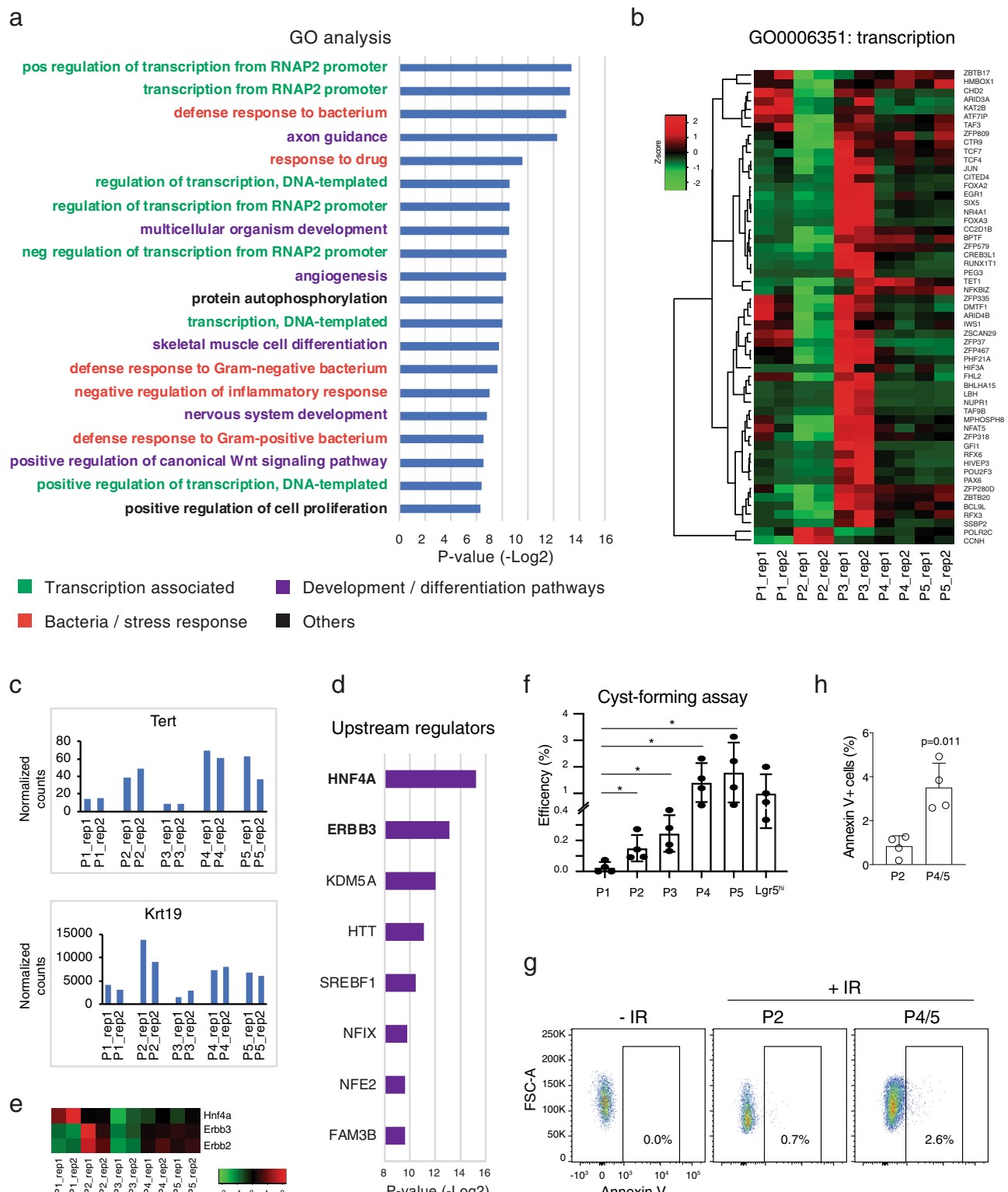

**Fig. 6 Characterization of the OSCAR intestinal P2 cell population. a** Bar chart of the -Log2 *p* values (adjusted) of the enrichment for the Gene Ontology (GO) terms found significantly enriched in the P2 OSCAR cell population. Adjusted enrichment *p* value is calculated using one-tail Fisher' exact test by DAVID (v6.8)[56]. **b** Hierarchical clustering and heatmap of the expression level of the differentially expressed genes (DEGs) found in the GO term "transcription from DNA template" in the indicated OSCAR samples. **c** Bar charts showing the RNAseq normalized counts levels of the Tert and Krt19 gene markers in the analyzed OSCAR populations. **d** Bar chart of the -Log2 *p* values of the enrichment for the upstream regulators of the upregulated genes in the P2 population with respect to the average of the other populations. The *p* value is calculated by one-tail Fisher's exact test by IPA (Ingenuity Pathway Analysis)[57]. **e** Heatmap of the expression level of Hnf4a and Erbb2/3 genes in the indicated OSCAR samples. **f** Bar chart indicating the cyst-forming efficiency of the indicated OSCAR populations in vitro. Number of cysts have been counted at day 5 after single-cell plating. $n = 4$ mice were analyzed. Data are presented as mean value ± SD. Paired two-tailed *t*-test was used. *$p$ value < 0.05 (p1 vs p2, *p* value = 0.0246, p1 vs p3, *p* value = 0.0138, p1 vs p4, *p* value = 0.0293, p1 vs p5, *p* value = 0.0354). Single Lgr5$^{high}$ cells isolated from Lgr5-eGFP mouse model have been used as control. **g** FACS analysis performed by using Annexin V staining to measure apoptosis in crypt cells from irradiated (+IR) mice. Left panel (−IR) shows FACS profile of ungated OSCAR cells (all the cells) isolated from intestinal crypt of not irradiated OSCAR mice. **h** Quantification of the Annexin V+ cells in irradiated OSCAR mice. $n = 4$ mice were analyzed. Data are presented as mean value ± SD. Welch's two-tailed *t*-test was used for the statistical analysis. Source data are provided as a Source Data file.

---

(150 V for 1.5 h). FLAG epitope was detected after blocking the membrane in 2% skimmed milk in PBS using mouse monoclonal anti-FLAG M2 antibody (Sigma-Aldrich F1804) 1:500 in blocking solution.

**Cell lines and lentivirus production**. Lentivirus was produced using plasmids CSII-EF Venus-P2A3F-mCherry, CSII-EF mCherry-P2A3F-Venus (control plasmids), or same plasmids with insertion of NPKATPPQI at various positions in the second fluorescent protein (experimental plasmids). Other plasmids used for lentivirus production were pCD/NL-BH*DDD, a gift from Jakob Reiser (Addgene plasmid # 17531), and pMD2.G, a gift from Didier Trono (Addgene plasmid # 12259). Lentivirus was produced by lipofection of HEK293T cells maintained in 37 °C / 5% CO$_2$ in a humidified incubator and infectious supernatant was harvested after 48 h.

The human melanoma cell lines 501 and Skmel-28 were maintained in RPMI 1640 medium (Life Technologies) with 10% FCS and Penicillin/Streptomycin at 37 °C / 5% CO$_2$ in a humidified incubator. For screening of Venus-NPKATPPQI insertion mutants, infectious supernatant was added to an MOI of 0.3. Flavopiridol (Sigma) was added at the indicated concentrations and the ratio Venus/mCherry determined by FACS analysis after 48 h.

Mouse small intestinal crypt organoids were derived from WT C57BL/6 mice after cervical dislocation and cultured according to Mahe et al.[51]. Briefly, the small intestine was surgically removed, flushed with cold PBS and incubated in crypt chelation buffer (2 mM EDTA in PBS) and subsequently in crypt dissociation buffer (2 g D-Sorbitol and 3 g Sucrose in 200 ml PBS). The crypt suspension was washed with PBS and resuspended in growth factor reduced Matrigel (Corning). Crypt culture medium consisted of Advanced DMEM/F12 (Invitrogen) supplemented with 10 mM HEPES, GlutaMAX, Penicillin/Streptomycin, N2 and B27 Supplement (all from Life Technologies). Furthermore, human recombinant Noggin (100 ng/ml), mouse recombinant R-Spondin (1 μg/ml) and human recombinant EGF (50 ng/ml, all from R&D Systems) were added to the culture. Lentivirus transduction was performed after mechanical dissociation of organoids, one wash with ice-cold PBS and resuspension in infectious supernatant containing 10 μM Y27623 and 8 μg/ml Polybrene (both from Sigma). Resuspended cells were centrifuged with infectious supernatant at 600*g* for 1 h at RT and incubated for 3–5 h at 37 °C in a humidified incubator. Cells were then washed with ice-cold PBS and resuspended in Matrigel for culture as above.

**Generation of the OSCAR mouse line**. The Rosa26 targeting vector R26TV-CAG-IRES-eGFP (a kind gift from Yoshiteru Sasaki) was digested with AscI and XmaI, releasing the CAG-IRES-eGFP cassette. mCherry-P2A3F-VeN90 was cloned by PCR containing AscI and XmaI sites and ligated into the vector. Next, the resulting vector was digested with PacI, releasing the CAG promoter, and a PacI-EF1a promoter-PacI cassette, created by PCR, was ligated. PCR conditions were 95 °C for 3 min, then 30 cycles of 95 °C for 30 s, 58 °C for 10 s, 72 °C for 1 min; a final extension at 72 °C for 3 min and hold at 4 °C. Correct identity of the plasmid was validated by sequencing. E14.1 ES cells were cultured on irradiated MEFs in the presence of LIF, sodium pyruvate, non-essential amino acids, L-Glutamine, Penicillin/Streptomycin, and beta-mercaptoethanol in DMEM supplemented with 15% FCS. ES cells in logarithmic growth phase were electroporated with 1 μg of R26TV-EF1a-loxP-NeoR-pA-loxP-OSCAR and further cultured for 48 h before selection with 25 μg/ml G-418. Individual clones were picked and further expanded in 96-well plates. Genomic DNA was extracted and PCR with primers spanning both the short and the long arm of the targeting vector performed, to confirm accurate integration of the knock-in. Furthermore, a plasmid carrying a *Cre* recombinase was transfected into properly targeted clones and expression of the OSCAR construct confirmed by FACS. Correct ES cell clones were then sent on dry

ice to PolyGene (www.polygene.ch), where blastocyst injection and transplantation to foster mothers were performed. Founder chimeric male mice were shipped back to Jena and bred with female Vasa-cre$^{tg/+}$ mice to remove the stop cassette ubiquitously. Other F0 founders were backcrossed to WT mice, maintaining the lox-stop-lox (lsl) cassette to allow lineage-specific expression of the OSCAR construct. Both mouse lines, EF1a-OSCAR and EF1a-lsl-OSCAR are available upon request.

We affirm that we have complied with all relevant ethical regulations for animal testing and research. We confirm that the study involving animals has been approved by the local competent authorities (THÜRINGER LANDESAMT FÜR VERBRAUCHERSCHUTZ) under the License FLI-17-028 and by the Animal Welfare Office at FLI institute.

**Mouse husbandry**. Mice were kept at an ambient temperature of 22 ± 2 °C, relative humidity of 55 ± 15% and a day/night cycle of 12 h/12 h. Individually ventilated cages (IVCs) were used. Cages and consumables were autoclaved before entering the animal houses.

**RNAseq analysis**. Small intestinal crypts of EF1a-OSCAR mice were isolated and a single cell suspension was derived by a 20 min incubation with TrypLE at RT. Cells were stained with rat monoclonal anti-EpCAM (CD326) PE-Cy7 (G8.8) (eBioscience 25-5791-80) at 1:100 dilution for 15 min on ice before washing and staining with Sytox Blue dead cell stain. FACS sorting was performed on a BD FACS ARIA III. 10,000 cells of each population were sorted and RNA was extracted using TRIzol Reagent. Concentration and integrity of isolated RNA was determined using Qubit 3.0 Fluorometer (Thermo Fisher) and Fragment Analyzer (Agilent). Poly-A library preparation was performed using TruSeq RNA Library Prep Kit v2 (Illumina) according to manufacturer's protocol. Briefly, poly-A containing mRNA was enriched using magnetic beads, first-strand synthesis of cDNA was done using SuperScript III Reverse Transcriptase (Thermo Fisher), second strand synthesis was performed using the second strand master mix (TruSeq RNA Library Prep Kit v2 (Illumina)) and the DNA was cleaned up using Agencourt AMPure XP Beads (Beckman Coulter). After end repair and dA-tailing, adaptors were ligated and the library was further enriched using the PCR Master Mix (TruSeq RNA Library Prep Kit v2 (Illumina)). Concentration and quality of the library were determined using Qubit 3.0 Fluorometer (Thermo Fisher) and Fragment Analyzer (Agilent). The libraries were pooled and 20pM loaded into the NextSeq 500 (Illumina) Sequencer using the NextSeq 500/550 High Output Kit v2.5 (75 cycles).

Fastq files quality check was performed using FastQC v0.11.5. The fastq files were mapped to the mm9 genome using TopHat v2.1.0[52] with the following parameters–bowtie1–no-coverage-search -a 5. The number of reads covered by each gene is calculated by HTSeq-Count 0.11.2[53] with -s no -a 0 -t exon -m intersection-nonempty parameters. Before further analysis, all of the rRNA genes were removed from the count data. For calculating differentially expressed genes and normalized count, DESeq2 R package v1.20.0[54] was used with the default parameters. For correlation analysis, PCA and plotting the expression, the normalized count was used. For cell cycle prediction, normalized count was analyzed by cyclone function in scran R package v1.12.1[55] using mouse_cycle_markers.rds file (from scran package) as the reference gene pairs. For geneset enrichment analysis, normalized counts (for each gene in all of the samples) were scaled using scale function in R (with center = TRUE, scale = TRUE parameters). The average of scaled data was calculated for each group and used for drawing plot. For Geneset enrichment analysis the following genesets were used: intestinal stem cells, enterocytes, Paneth cells, Goblet cells, Tuft cells, Enteroendocrine (from[39]); secretory precursors (from[41]); enterocyte progenitors, and transit-amplifying cells (in-house generated datasets from scRNAseq

experiment - available upon request). GO analysis was done by uploading differentially expressed gene list (DESeq2 results, adjusted $p$ value < 0.01) in DAVID database (v6.8)[56]. For identification of potential upstream regulators, differentially upregulated genes (DESeq2 results, $p$ value < 0.05, log2 fold change > 0) were uploaded in IPA (Ingenuity Pathway Analysis) v45868156[57].

**Fluorescence-activated cell sorting (FACS) analysis and organoid-forming assay.** For single cells preparation, isolated crypts were dissociated in TrypLE containing Y-27632 (20 μM), Thiazovivin (10 μM), and CHIR99021 (2.5 μM) at 37 °C for 20 min. Cells were washed and stained with rat monoclonal anti-EpCAM (CD326) PE-Cy7 conjugated (G8.8) antibody (eBioscience 25-5791-80) at 1:100 dilution for 30 min on ice. After washing, cells were resuspended in 2% FBS in PBS supplemented with Y-27632 (20 μM), Thiazovivin (10 μM), and CHIR99021 (2.5 μM). Sytox blue was used for live-cell separation and different cell populations were sorted using a FACSAriaII (BD Biosciences). Sorted cells were resuspended in Matrigel supplemented with additional EGF (100 ng/ml), Noggin (100 ng/ml), R-spondin-1 (500 ng/ml), Wnt3A (100 ng/ml), Jagged-1 peptide (10 μM), Y-27632 (20 μM), Thiazovivin (10 μM), and CHIR99021 (2.5 μM). The cells were overlaid with Advanced DMEM/F12 containing B27 and N2 supplement, EGF (100 ng/ml), Noggin (100 ng/ml), R-spondin-1 (500 ng/ml), and N-acetylcystein (1.25 mM). Colony-forming capacity was quantified on day 5 by counting the number of cysts under microscope. For Hoechst staining, after single cell preparation (see above), cells were washed and stained with EpCAM antibody at 1:100 dilution in 2% FBS in PBS supplemented with Y-27632 (20 μM), Thiazovivin (10 μM), and CHIR99021 (2.5 μM) for 30 min on ice. After washing, cells were incubated with Hoechst 33342 (1:1000) in 2% FBS in PBS supplemented with Y-27632 (20 μM), Thiazovivin (10 μM), and CHIR99021 (2.5 μM) for 30 min at 37 °C, washed and analyzed by flow cytometry on UV laser-equipped FACSAria Fusion (BD Biosciences). For Annexin V staining, mice were irradiated with 10 gray and intestinal tissue were isolated from mice after 6 h and processed for single cells preparation as above; following EpCAM staining for 30 min on ice, cells were washed and resuspended in 1 ml of 1X Binding Buffer from Annexin V kit. Then 5 μl of APC Annexin V were added to the suspension and cells were placed at RT for 15 min. After washing with 1X Binding Buffer once, cells were analyzed using FACSAriaII (BD Biosciences). Quantification and statistical analysis were performed by using PRISM v.8 as indicated in the legend of each analysis.

**RNA isolation from the OSCAR cell populations and RT-qPCR.** After FACS sorting, the sorted cells of each population (average of ~15,000 cells per population) were centrifuged at 500*g* for 5 min at 4 °C and the total RNA was isolated using Quick-DNA/RNA™ Microprep Plus Kit (Zymo Research) according to the manufactures' protocol. Isolated RNA was quantified on Qubit 3.0 (Thermo Fisher Scientific). For analysis in Fig. 5h, RNA amount of all the populations from three independent experiments was normalized on the number of cells and on replicate 1 of P1 population to show relative fold change. qRT-PCR was performed by using SuperScript™ III Platinum™ One-Step qRT-PCR Kit (Thermo Fisher Scientific). Quantification and statistical analysis were performed by using PRISM v.8 as indicated in the legend of each experiment.

**Graphical representation of the analysis.** All the bar charts indicate the average of the values of the single samples and the error bars indicate the standard deviation (SD) value. All the boxplots indicate median with interquartile range distribution of the values.

**Reporting summary.** Further information on research design is available in the Nature Research Reporting Summary linked to this article.

## Data availability
Sequencing data are deposited on GeoDatasets under the Accession Number GSE147030. Source data are provided with this paper.

## Code availability
Custom R scripts can be found on github at https://github.com/smmrasa/OSCAR.

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

## Acknowledgements

This work was supported by funding from the Ludwig Institute for Cancer Research, NIH grant PO1 CA128814-06A1, Alexander von Humboldt foundation (1164767-ITA-SKP), The Wellcome Trust Institute, DFG Grant NE 2144/5-1 and the Leibniz Institute on Aging (FLI). The authors would like to thank Pravin Mahajan and Sarah Picaud for help with protein purification and characterization, Marc Fisher for support with microscopy and Joanna Kirkpatrick for Mass Spec analysis. The authors are grateful for the support by core facilities FACS, Animal Facility, and Microscopy at the Leibniz Institute on Aging - Fritz Lipmann Institute.

## Author contributions

R.F. performed most of the experiments with the contribution of K.H., P.F., S.K., F.A., A.M., A.K., and O.O.; F.A. and L.A. performed RNAseq. M.R. and F.N. analyzed RNAseq data. Z.Q.W. and T.R. provided support for mouse transgenesis. R.F., C.R.G., and F.N. designed the study, discussed the results, and wrote the manuscript with input from all the authors.

## Funding

## Competing interests

The authors declare no competing interests.
