## [Peer Review File · Nature Communications]

Reviewers' Comments:

Reviewer #1:

Remarks to the Author:

Review of Establishment of a novel fluorescent reporter of RNA-polymerase II activity to identify dormant cells by Freter R et al.

This paper utilizes a novel fluorescent readout, named OSCAR, that acts as a proxy for RNA Pol2 activity, to identify cells that are transcriptionally dormant.

Issues:

1) The introduction makes an attempt to distinguish features of quiescence and dormancy. It's not clear to me how the authors conclude that dormant cells are considered to have low metabolic activity and global transcriptional levels relative to quiescent cells. Dormancy is a vague, non-scientific term that has little precise meaning in the literature. Quiescence, on the other hand, has a more specific definition, defined largely in the stem cell compartments of the muscle, blood, and intestine. In contrast to the statements in the introduction, Quiescence, frequently associated with being in the G0 stage of the cell cycle, implies several things: metabolic dormancy (low mTORc1), being non-proliferative but retaining the capacity to re-enter the cell cycle, low levels of global transcription, a high nuclear-to-cytoplasmic ration, and, depending on the organ system, the expression of quiescence-maintaining transcriptional regulators. Essentially, the authors are using the term 'dormancy' to describe properties associated with quiescent stem cells.

The authors cite a couple of obscure papers to support their argument that the properties traditionally associated with quiescence are really associated with 'dormancy'. This is an unusual argument given that the dictionary definition of quiescent and dormant are the same. The authors should consider refining their semantics and present a more representative discussion of what 'quiescence' means (e.g., see review by Rando <https://www.nature.com/articles/nrm3591>). Also see: PMC4015626 , PMC4065227 , PMC4961601 , among many, many others.

If the authors are trying to distinguish between quiescent and metabolically active cells, the difference is best characterized in the literature by distinct cell cycle states: G0 being quiescence as discussed above, and G1 arrested being non-cycling cells that are metabolically and transcriptionally active. For example, in the intestinal system in which the authors are working, cells arrested in G1 are terminally differentiated secretory cells and enterocytes, relative to cells in G0, and the two are distinguished by metabolic or global transcriptional levels (in particular, the Hoechst/pyronin assays has been widely applied in both the intestinal and hematopoietic stem cell systems to distinguish cells in G0 from G1 based on their global RNA content).

2) With respect to the intestinal stem cell literature in particular, the discussion about Bmi1 and Hopx is not well fleshed out- Bmi1 and Hopx mRNA can be found in Lgr5 cells, but Bmi1-CreER and Hopx-CreER mark quiescent cells that are Lgr5-negative. So the authors should be specific about this point as it has caused much confusion in the literature. Lrig1 is even more confusing, as the CreER driver is discordant with the RNA which may or may not be concordant with the antibody, depending on which antibody you use. See PMC4235148 for discussion about these discrepancies. Similarly, single cell transcriptomic profiling (see PMC6022292) shows that transcripts from Prox1, Dll1, and Krt19 are not really specific for any particular population (Prox1 and Dll1 are enriched, but not specific to the secretory lineage, Krt19 marks essentially every cell above the crypt base so is of little value when thinking about cell type-specific functions)

3) Figure1C- in the intestine, how do we know these are epithelial cells? It would be nice to show a membrane stain like E-cadherin

4) Fig 2E-H is hard to interpret based on the low resolution

5) The transcriptome analysis is interesting, but given that we now have many single cell RNAseq datasets that don't reveal a population consistent with 'P2' is concerning. Ideally the authors would perform scRNAseq and relate this to published data

Similarly, Tert gene expression and enzymatic activity has clearly been shown to be highest in Lgr5+ cells, and, as with many other markers of reserve ISCs, Tert-GFP/Cre does not seem to correlate with Tert enzymatic activity (see PMC3061032)

6) The organoid formation assays in the final figure are a bit confusing also- 'reserve' stem cells isolated by Bmi or Hopx-CreER have much higher organoid forming capacity than differentiated cells (which should have very little), but have less than Lgr5+ cells, the gold standard in these assays. This assay should include Lgr5+ cells as a reference, and images of these cultures should be shown.

7) Quiescent cells in vivo cannot be maintained in this state in vitro, a longstanding problem in the hematopoietic stem cell field and something also observed in the ISC field- once taken from the niche and exposed to the nutrient/cytokine milieu, they exit dormancy (quiescence). This it would be interesting to isolate OSCAR high cells from mice and follow the activity of the OSCAR reporter over time.

Overall, this is a really neat tool, and while I'm not an expert in the technical background leading to the generation of this tool, the data in Figure 2 are pretty compelling. The paper falls a bit short on presenting convincing evidence that this tool identifies the elusive reserve intestinal stem cell. To build this case, the authors could employ several approaches including scRNAseq, but more importantly demonstrate that OSCAR high cells exhibit the most important hallmark of this population: resistance to high levels of DNA damage followed by cell cycle entry and regeneration. One very compelling experiment would be irradiate mice with high doses of gamma radiation (10-14Gy), where reserve/quiescent stem cells are known to survive, while Lgr5 cell cannot. Shortly after irradiation, prior to initial mitosis for regeneration or tissue destruction (2hrs post-IR), sort cells as shown, then stain for markers of apoptosis. In this assay, the dormant ISCs should be resistant to apoptosis in contrast to actively cycling cells. Terminally differentiated cells would also be resistant, but these cells can easily be distinguished based on their high expression of lineage-specific genes.

Taken together, this is a great tool, but the application in the intestine and biological conclusion needs some work.

Reviewer #2:

Remarks to the Author:

The authors have developed an interesting strategy to identify and isolate live dormant cells based on the notion that phosphorylation of RNA Polymerase II (RNAPII) is largely absent in dormant cells. By combining an engineered nuclear Venus reporter that is sensitive to CDK9 kinase activity, and an mCherry reporter that is not, they generated a ratiometric fluorescent Optical Stem Cell Activity Reporter (OSCAR) system to report overall transcriptional activity and to identify cells with low levels of CDK9 activity. They then applied this system first to intestinal organoids and then to intestinal crypts in vivo, using a newly generated ROSA26 OSCAR reporter mouse. They conclude that this model has allowed for the identification and isolation of several populations of transcriptionally diverse OSCAR^{high} and OSCAR^{low} intestinal epithelial cell states in vivo. They also argue that they could distinguish between dormant "reserve" stem-like cells and quiescent but activated secretory lineage precursor cells in the small intestine. In general this is an interesting and thoughtful study, but several concerns are highlighted below.

Major Comments:

It would be helpful visualize the staining pattern of RNAPII-pSer2 throughout the crypt and at

crypt villi boundary in Figure 1 C.

In Figure 3F/G, please provide representative pictures of Ki67 staining.

Regarding Figure 2a, is there a population that is predicted to be yellow with both high Venus and mCherry expression? I ask because the subsequent data in Figure 2b appear to show this. If dormant cells express low levels of mRNA, as proposed by the model, then why would the mCherry signal in the 2 cells marked by open arrowed in Figure 2B be among the brightest of cells?

The inserts for Figure 2 E-H do not appear to be the same optical section as the phase contrast image. There is likely a technical explanation. Please provide these details.

Also, the bottom phase images are not necessary and waste space. It would be far better to show higher power views of the data which is hard to see. Especially the 3 green cells at the 42-hour time point.

The "data not shown" in line 230 need to be shown. This would address the point raised above.

A merged image of Figure 3A would help see the 3 cells we're meant to focus on. Also, there appear to be other cells that could have been labeled with arrows, that were not. Please also use arrows in the DAPI channel. On line 250 the "OSCAR^{high} (Ven90^{high} mCherry^{low}) cells the claim about cells being mCherry^{low} is subtle and very difficult to see, at least for this reviewer. One wonders if the images are overly saturated. Can the fluorescent data be presented in a complementary way to make the differences clearer visually? Also, the authors should consider including the raw data for each cell analyzed as supplemental information. The number of independent experiments and animals should also be included in the legend, not simply the number of cells analyzed.

The authors note that the small intestine of EF1a-OSCAR mice was fixed in 4% PFA for only 20 min at room temperature because the endogenous OSCAR fluorescence is very sensitive to fixation. Because the analysis is done using PFA fixed frozen sections, there should be a discussion of how tissue autofluorescence was handled. In addition, because all of the quantitative data presented in Figure 3b-g are dependent on the measurement of endogenous fluorescence of individual cells, a more detailed analysis of how these data were collected, and how the OSCAR ratio was calculated for each cell, is essential for the reader to understand the strengths and limitations of the data. As presented it would be difficult for another group to repeat these studies.

Figure 4a and b show the relative expression of Ven90 and mCherry using flow cytometry. I find it curious why mCherry would be expressed over a larger dynamic range (~2.5 logs) than Ven90 (~1.5 logs). Also, the diagonal nature of the data raise questions about whether some of the signal may be explained by autofluorescence.

The authors should show the complete set up of the FACS experiments including proper color controls (unstained, single colors, fluorescence minus one) and include the viability staining as well as FSC/SCC and doublet analysis, etc. Single color controls will require separate mice expressing mCherry or Ven90 alone, which are necessary to properly set gates and allow for compensation, etc. Wildtype organoids should be used for the unstained, viability and EpCAM controls.

On line 894 the figure legend states that "Population P1 and P3 contain OSCAR^{low} cells, while P2 contains OSCAR^{high} cells." The fluorescence intensity of Ven90 for P2 and P3 are essentially the same. What distinguishes them better is their mCherry expression (high-medium-low).

Regarding the RNA-seq data presented in Figure 4 and 5, it is understandable why the authors would rely on 2 biological replicates. However, these data should be considered as screening data and not overly interpreted at the single gene level, as was done in 5b. Rather, conclusions about

single genes should be performed and validated on independently sorted samples with larger replicate numbers using qRT-PCR analysis.

Consistent with the prior commentxx , it is not OK to make statements such as this on line 305: "Remarkably, the P2 population did not show any enrichment of these known markers, except a slight enrichment of stem cell markers (Olfm4 and Ascl2) in one of the two replicates (Figure 5B, top-left panels)." This is the equivalent of showing an n=1 out of 2 samples analyzed.

It would be interesting to see what would happen to the OSCAR populations in an injury model (e.g. radiation, when Lgr5 cells are depleted/damaged).

As above, the gene expression data for Tert and Krt19 in the P2 population presented in Figure 6C should be performed and validated on independent samples.

Minor comments:

Regarding the observation that TERT/GFP cells are rare, I believe that the original description of 1 cell per 150 crypts in REF 10 was based on tissue sectioning. From my reading of REF 11 it would appear that this conclusion has been updated using whole crypt analysis to reveal that the majority of crypts have 1 or 2 cells positive cells.

Lines 187-188 are awkward and hard to understand.

Please provide split channel images for Figure 1A so the nuclear staining can be seen clearly.

In Figure 1C please define what the filled arrowhead refers to.

In Figure 3F/G, please provide representative pictures of Ki67 staining to accompany the figure.

On Line 161 remove reference to Figure S2A.

On Page 5 the authors define use in vivo when analyzing human melanoma cell lines. Please correct.

The reference in line 80 needs to be fixed.

Reviewer #3:

Remarks to the Author:

This manuscript describes OSCAR, an innovative approach to identifying cells with low transcriptional activity. The authors have also created a mouse expressing OSCAR in all cells. The cells demonstrated to have high OSCAR/low transcriptional activity do not belong to identifiable differentiated cells of the intestine and the authors propose that these cells are dormant stem cells. There remain several questions that the authors should address in a revised manuscript.

1) During the cell cycle, transcription decreases drastically during mitosis. Are the cells that were identified by the authors as dormant actually undergoing mitosis, with this stage of the cell cycle being the reason for lower transcriptional activity rather than the proposed dormancy? Can the authors distinguish low transcriptionally active mitotic cells from dormant cells?

2) LGR5 is reported by the authors to marks highly cycling stem cells. Data for the expression level of this gene should be reported in figs. 5 and 6 along with genes that identify other intestinal crypt cell types.

Minor comments

In a revised version of this manuscript, the authors should correct the reference citation on line 396 and make more consistent the reference throughout the manuscript to the OSCAR fluorescence ratio, which is variably referred to as OSCAR , OSCAR ratio fluorescence and OSCAR fluorescence throughout the manuscript. On line 877, the statement "while the 10% of OSCAR^{low} cells...." is unclear and should be corrected.

POINT-BY-POINT RESPONSE TO THE REVIEWERS' COMMENTS

We very much appreciated the thoughtful work of the reviewers and have revised the manuscript to address fully their comments. We note that referee1 regarded the OSCAR reporter as a 'neat' tool, referee 2 states that 'this is an interesting and thoughtful study' and referee 3 states that we take an 'an innovative approach to identifying cells with low transcriptional activity'. We provide below a point-by-point response to the reviewer's comments.

REVIEWER COMMENTS

Reviewer #1 (Remarks to the Author):

Review of Establishment of a novel fluorescent reporter of RNA-polymerase II activity to identify dormant cells by Freter R et al. This paper utilizes a novel fluorescent readout, named OSCAR, that acts as a proxy for RNA Pol2 activity, to identify cells that are transcriptionally dormant.

Issues:

1) The introduction makes an attempt to distinguish features of quiescence and dormancy. It's not clear to me how the authors conclude that dormant cells are considered to have low metabolic activity and global transcriptional levels relative to quiescent cells. Dormancy is a vague, non-scientific term that has little precise meaning in the literature. Quiescence, on the other hand, has a more specific definition, defined largely in the stem cell compartments of the muscle, blood, and intestine. In contrast to the statements in the introduction, Quiescence, frequently associated with being in the G0 stage of the cell cycle, implies several things: metabolic dormancy (low mTorc1), being non-proliferative but retaining the capacity to re-enter the cell cycle, low levels of global transcription, a high nuclear-to-cytoplasmic ration, and, depending on the organ system, the expression of quiescence-maintaining transcriptional regulators Essentially, the authors are using the term 'dormancy' to describe properties associated with quiescent stem cells.

The authors cite a couple of obscure papers to support their argument that the properties traditionally associated with quiescence are really associated with 'dormancy'. This is an unusual argument given that the dictionary definition of quiescent and dormant are the same. The authors should consider refining their semantics and present a more representative discussion of what 'quiescence' means (e.g., see review by Rando <https://www.nature.com/articles/nrm3591>). Also see: PMC4015626, PMC4065227, PMC4961601, among many, many others. If the authors are trying to distinguish between quiescent and metabolically active cells, the difference is best characterized in the literature by distinct cell cycle states: G0 being quiescence as discussed above, and G1 arrested being non-cycling cells that are metabolically and transcriptionally active. For example, in the intestinal system in which the authors are working, cells arrested in G1 are terminally differentiated secretory cells and enterocytes, relative to cells in G0, and the two are distinguished by metabolic or global transcriptional levels (in particular, the Hoechst/pyronin assays has been widely applied in both the

intestinal and hematopoietic stem cell systems to distinguish cells in G0 from G1 based on their global RNA content).

Response of the authors:

We thank the reviewer for the extensive review and for her/his comments that helped us to identify the weak points in our study. We agree that Dormancy and Quiescence are terms that are important to define and that are often mis-used or one used instead of the other. We would like to point out the term quiescence is not only used in the stem cell field and/or only referring to stem cells. Hepatocytes, fibroblasts, endothelial cells and many other cells can be indeed quiescent and are defined as quiescent in many scientific studies (e.g. PMID: 30146488, PMID: 12490302, PMID: 21049082). These differentiated cell types are not stem cells, they do not exhibit low metabolic and transcriptional activity and, according to the Rando review (PMID: 23698583, see for example Box 1), they can be defined as being in a G0 state of the cell cycle. In other words, quiescence in stem cells and quiescence in other cell types are not the same, and indeed some stem cells, such as the Lgr5+ population in the intestine, are not quiescent. Given these premises, we don't feel confident in strictly defining quiescent cells as those cells exclusively in G0, and metabolically and transcriptionally inactive. We apologize that this was not clear enough in the introduction. As a consequence, we rephrased some of the sentences to clarify this point in the revised manuscript.

Nevertheless, we agree that when referring to stem cells the terms Dormancy and Quiescence are highly overlapping and often interchangeably used. However, adult stem cell pools have been demonstrated in the last years to be heterogeneous and composed of different sub-populations. The hematopoietic compartment is an example, where dormant and active HSCs have been detected within the conventional quiescent HSC fraction (PMID: 31851939). It is also possible to speculate that quiescent HSCs can be in G0, but having subpopulations with different metabolic and transcriptional activity levels. In this case, the term dormant stem cells identify those cells that are with the lower metabolic and transcriptional activity level. Another example is the muscle stem cells (MuSCs) that have been shown to be composed of two distinct functional phases: G0 and an "alert" phase termed G_{Alert} ; "G0-to- G_{Alert} transitions were observed in several populations of quiescent stem cells" (PMID: 24870234). Because of this heterogeneity, we think that is more appropriate to indicate stem cells that are strictly in G0, with low metabolic and transcriptional activity as dormant stem cells, but that these will, depending on the population examined, either represent a subset of quiescent stem cells or will completely overlap with quiescent stem cells. The term dormant stem cell is indeed very used in different stem compartments (PMID: 20182459, PMID: 26686631). We specified that somatic stem cells can be often defined as dormant stem cells or quiescent stem cells interchangeably in the revised manuscript.

In the intestinal compartment, slow cycling, injury-resistant reserve intestinal stem cells have been identified (beside the highly-proliferative Lgr5+ cells). These putative reserve ISC have been identified using a variety of methods, including transgenic reporter mice (Tert, Hopx, Bmi1, ect) and DNA label retention. Comparison of cells isolated with different methods revealed a strong heterogeneity among the reserve ISCs bearing differences at

transcriptional level and in the cell cycle state (G0/G1 arrest) (PMID: 27237597). In fact, in our opinion, defining and classifying the reserve stem cell pool in the intestinal is currently very challenging and we agree with the reviewer the need to refine the semantics throughout the revised manuscript.

We agree with the reviewer that our tool is able to distinguish between transcriptionally inactive and active cells, with 1) a subgroup of reserve ISCs being transcriptionally inactive (OSCAR high - P2 population), supposedly in G0, and 2) another subgroup of reserve ISCs (OSCAR low - P3 population), transcriptionally active and characterized by secretory lineage markers. These P3 cells are differentiated or semi-differentiated cells, not-cycling, with the potential to re-enter in the cell cycle and replace the ISCs; according to the reviewer in a G1 arrest, but according to our opinion and other studies (like the Rando review) they could be defined being in a G0 state. Similarly to other differentiated cells like fibroblasts, the P3 cells can be properly termed as quiescent cells, therefore we don't feel confident in defining this P3 population as not quiescent in contrast to the P2 population being quiescent. From our point of view both populations are quiescent, but the P2 cells are transcriptionally inactive and can be better defined as dormant cells.

To clarify these aspects, we did further experiments suggested by the reviewer. We performed Hoechst FACS cell cycle analysis (Figure 5f,g) in the revised manuscript. Unfortunately, the Pyronin Y assay was not successful since it is exactly at the mCherry wavelength, but we measured the relative RNA content of the different cell population after sorting (Figure 5h). The two experiments indicated that both the P2 and P3 populations contain cells in G1 (or G0) phase of the cell cycle (P2>98%; P3>95% in 4 different biological replicates). The RNA content analysis showed that P2 has ~5 fold less total RNA than the P3 population. These experiments confirmed that the P2 cells are cell-cycle paused cells and indicated they are less transcriptionally active and therefore can be better defined as dormant cells in contrast to the P3 population that are cell-cycle arrested, but transcriptionally active.

2) With respect to the intestinal stem cell literature in particular, the discussion about Bmi1 and Hopx is not well fleshed out- Bmi1 and Hopx mRNA can be found in Lgr5 cells, but Bmi1-CreER and Hopx-CreER mark quiescent cells that are Lgr5-negative. So the authors should be specific about this point as it has caused much confusion in the literature. Lrig1 is even more confusing, as the CreER driver is discordant with the RNA which may or may not be concordant with the antibody, depending on which antibody you use. See PMC4235148 for discussion about these discrepancies. Similarly, single cell transcriptomic profiling (see PMC6022292) shows that transcripts from Prox1, Dll1, and Krt19 are not really specific for any particular population (Prox1 and Dll1 are enriched, but not specific to the secretory lineage, Krt19 marks essentially every cell above the crypt base so is of little value when thinking about cell type-specific functions)

Response of the authors:

We agree that there are discrepancies in the literature regarding expression of markers for the identification of quiescent intestinal cells. We edited the Introduction, discussing more the difference between mRNA expression level and transgenic reporter activity of these markers (e.g. Bmi1 and Hopx) accordingly to the reviewer's suggestion (lines 67-70).

3) *Figure 1C- in the intestine, how do we know these are epithelial cells? It would be nice to show a membrane stain like E-cadherin*

Response of the authors:

Unfortunately, we did not perform E-Cadherin (or other epithelial cell marker) staining in that experiment. Instead, we repeated the experiment by adding EpCAM (marker of intestinal epithelial cells) staining and we added an immunofluorescence image in Figure 1c and S1c of the revised manuscript. The new experiment shows that RNAPII-pSer2 negative cells in the intestinal crypt are epithelial cells.

4) *Fig 2E-H is hard to interpret based on the low resolution*

Response of the authors:

The pictures are taken from the time-lapse movie (Supplementary movie 1) made with a 20x objective to avoid too strong light exposure. One of the problems with this assay is that organoids grow in all directions, and very often they go out of focus. We have tried to provide better resolution images and we added a magnification in Figure S3 of the revised manuscript. The referee can also view the movie itself.

5) *The transcriptome analysis is interesting, but given that we now have many single cell RNAseq datasets that don't reveal a population consistent with 'P2' is concerning. Ideally the authors would perform scRNAseq and relate this to published data. Similarly, Tert gene expression and enzymatic activity has clearly been shown to be highest in Lgr5+ cells, and, as with many other markers of reserve ISCs, Tert-GFP/Cre does not seem to correlate with Tert enzymatic activity (see PMC3061032)*

Response of the authors:

We agree with the reviewer that *'many single cell RNAseq datasets don't reveal a population consistent with "P2" is concerning'*. Along the same lines, it is also concerning, that these datasets do not reveal populations consistent with the mTert-GFP+ cells identified as dormant stem cells (PMID:21173232, PMID:26686631). Even more concerning is that these datasets do not clearly reveal the populations of slow-cycling Label-Retaining cells or any other populations previously reported as reserve intestinal stem cells (PMID:27237597, PMID:28686870, PMID:22190486, and others). One of the hypotheses may be that the reserve (and/or dormant) intestinal stem cell population is an heterogenous cell population composed of cells expressing markers of different lineages. These cells, may, therefore, cluster with the differentiated or precursor cells of that

specific lineage more than cluster together as a reserve intestinal stem cell cluster. The "reserve" population may therefore be considered more a "state" than a specific cell-type, and as a consequence be difficult to transcriptionally profile and cluster. Cells of the enteroendocrine and of the enterocyte lineages, Paneth cells, secretory precursors have been demonstrated to be able to dedifferentiate and replace Lgr5+ under specific circumstances (PMID:28686870, 28648363, 29887318, 30157426, 26831517) demonstrating the high plasticity of the intestinal epithelium. It is tempting to speculate whether the usage of specific cell marker-based reporter together with lineage tracing experiment is a sufficient method to define reserve ISCs.

Another hypothesis, that we discussed in the "Discussion" section of the manuscript, may be that dormant cells, having an absence of (or at least very low) productive mRNA transcription, may fail to be "captured" during the polyA-RNA enrichment step or to pass the downstream quality filters used in scRNAseq experiment or analysis. For example, in these studies (PMID:30392957, PMID: 29144463) all epithelial cells with fewer than 800 detected genes were excluded or in this other study (PMID: 32351704) cells with fewer than 200 genes, with UMI count lower than 1000 or with mitochondrial gene percentage higher than 10% were excluded. Dormant cells, identified by the OSCAR construct, have low RNAP2-mediated transcription therefore these cells have a low amount of mRNA (and consequently low number of expressed genes) suggesting that they may not pass these quality filters.

6) The organoid formation assays in the final figure are a bit confusing also- 'reserve' stem cells isolated by Bmi or Hopx-CreER have much higher organoid forming capacity than differentiated cells (which should have very little), but have less than Lgr5+ cells, the gold standard in these assays. This assay should include Lgr5+ cells as a reference, and images of these cultures should be shown.

Response of the authors:

We agree with the reviewer that the Lgr5+ cells are the gold standard and we actually use them as control in these assays. We have included the Lgr5-high cells in the cyst-forming assay in the revised manuscript (Figure 6f). They indeed form more cysts than the P2 and P3 populations. We have also added pictures of these cultures in the revised manuscript (Figure S6).

7) Quiescent cells in vivo cannot be maintained in this state in vitro, a longstanding problem in the hematopoietic stem cell field and something also observed in the ISC field- once taken from the niche and exposed to the nutrient/cytokine milieu, they exit dormancy (quiescence). This it would be interesting to isolate OSCAR high cells from mice and follow the activity of the OSCAR reporter over time.

Response of the authors:

We agree with the reviewer that quiescent cells cannot be maintained in this state *in vitro* and that it would be interesting to follow the OSCAR activity over the time from a single cell. However, most of the P2 cells die when put in culture (organoid forming efficiency around 0.15%) and we failed to successfully do time-lapse microscopy from a single cell. At the cyst stage (where it is possible to identify the survivor cells) the P2 cells already resemble a intestinal cyst in culture and most of the cells are OSCAR low/medium (more red/yellow than green - see figure here below). We showed the dynamics of the OSCAR construct in organoid culture in the Supplementary video 1.

Figure here shows a cyst (arrowhead) originating from P2 cell. Most of the cells in the cyst are red or yellow indicating that they already did exit from dormancy. Arrows indicate instead two P2 cells that didn't develop cyst (potentially dying/dead cells or cells that have remained 'dormant').

Overall, this is a really neat tool, and while I'm not an expert in the technical background leading to the generation of this tool, the data in Figure 2 are pretty compelling. The paper falls a bit short on presenting convincing evidence that this tool identifies the elusive reserve intestinal stem cell. To build this case, the authors could employ several approaches including scRNAseq, but more importantly demonstrate that OSCAR high cells exhibit the most important hallmark of this population: resistance to high levels of DNA damage followed by cell cycle entry and regeneration. One very compelling experiment would be irradiate mice with high doses of gamma radiation (10-14Gy), where reserve/quiescent stem cells are known to survive, while Lgr5 cell cannot. Shortly after irradiation, prior to initial mitosis for regeneration or tissue destruction (2hrs post-IR), sort cells as shown, then stain for markers of apoptosis. In this assay, the dormant ISCs should be resistant to apoptosis in contrast to actively cycling cells. Terminally differentiated cells would also be resistant, but these cells can easily be distinguished based on their high expression of lineage-specific genes. Taken together, this is a great tool, but the application in the intestine and biological conclusion needs some work.

Response of the authors:

We agree with the reviewer that resistance to high levels of DNA damage followed by cell cycle entry and regeneration represents an hallmark of the reserve stem cell population. To address this point, we performed total body gamma irradiation on OSCAR mice and measured the apoptosis marker Annexin V by FACS analysis. We found that P4/P5 cells (putative ISCs) have a higher percentage of AnnexinV+ cells than P2 cells indicating that the OSCAR^{high} cells are more resistant to DNA damage following irradiation (Figure 6h of the revised manuscript).

Reviewer #2 (Remarks to the Author):

The authors have developed an interesting strategy to identify and isolate live dormant cells based on the notion that phosphorylation of RNA Polymerase II (RNAPII) is largely absent in dormant cells. By combining an engineered nuclear Venus reporter that is sensitive to CDK9 kinase activity, and an mCherry reporter that is not, they generated a ratiometric fluorescent Optical Stem Cell Activity Reporter (OSCAR) system to report overall transcriptional activity and to identify cells with low levels of CDK9 activity. They then applied this system first to intestinal organoids and then to intestinal crypts in vivo, using a newly generated ROSA26 OSCAR reporter mouse. They conclude that this model has allowed for the identification and isolation of several populations of transcriptionally diverse OSCAR^{high} and OSCAR^{low} intestinal epithelial cell states in vivo. They also argue that they could distinguish between dormant “reserve” stem-like cells and quiescent but activated secretory lineage precursor cells in the small intestine. In general this is an interesting and thoughtful study, but several concerns are highlighted below.

Response of the authors:

We thank the reviewer for considering our study interesting and thoughtful and for the comments and concerns that she/he has raised up. We think that these weak points were important to be addressed and thanks to the implemented analysis and experiments, our study has greatly improved. Here below, a point-by-point response to the major and minor concerns.

Major Comments:

1) It would be helpful visualize the staining pattern of RNAPII-pSer2 throughout the crypt and at crypt villi boundary in Figure 1 C.

Response of the authors:

We agree with the reviewer that staining of RNAPII-pSer2 at crypt-villi boundary is very helpful. We added full immunofluorescence images (both single colors and merge) in Figure S1c of the revised manuscript. The new pictures show cells in the intestinal crypts, in crypt-villi boundary and in the beginning of the villus.

2) In Figure 3F/G, please provide representative pictures of Ki67 staining.

Response of the authors:

Following to the reviewer suggestion, we have added representative image of Ki67 staining on an intestinal crypt of OSCAR mice in Figure S4a.

3) Regarding Figure 2a, is there a population that is predicted to be yellow with both high Venus and mCherry expression? I ask because the subsequent data in Figure 2b appear to show this. If dormant cells express low levels of mRNA, as proposed by the model, then why would the mCherry signal in the 2 cells marked by open arrowed in Figure 2B be among the brightest of cells?

Response of the authors:

Yes, some cells are supposed to be yellow, specifically those that are switching between a low and a high transcriptional activity. In the Supplementary Video that shows a time lapse imaging of OSCAR organoids, it is possible to see the dynamics of colors conversion.

Figure 2b-c shows a fully formed organoid bud (crypt-like structure). The inter-cell differences in the mCherry signal here may be due to the fact that this organoid has been transduced with lentivirus containing the OSCAR construct. In transduced organoids, it is difficult to have the same level of transgene delivery and expression. Above-right the two cells with empty arrows, there is indeed a group of cells that express much less of the OSCAR construct (Low mCherry signal). Therefore, it is not appropriate to compare different cells without an intra-cell normalization of the signal. The two filled arrows indicate two cells that express almost the same level of mCherry, whereas the differential signal of the VeN90 protein is much higher. The VeN90 signal has to be always normalized to its same-cell mCherry control. When we calculated this ratio (OSCAR ratio) we were able to observe an inverse correlation between this ratio and RNAPII-Ser2 signal (Figure 2d). This further confirmation of the functionality of the OSCAR reporter pushed us to generate a transgenic mouse with OSCAR that also enabled us to reduce the issues of heterogenous expression arising from the lentivirus-based transduction-derived transgene. The OSCAR mouse turned out to be a much more reliable system, where we actually found the very same observation (Figure 3d). The single-channel fluorescent signals are much more reliable in the OSCAR mouse system (e.g. Fig 3a), while in the transduced cell (due to the transgene expression variability) we can only feel confident in drawing conclusions from the fluorescent ratio (Venus/mCherry) signal.

4) The inserts for Figure 2 E-H do not appear to be the same optical section as the phase contrast image. There is likely a technical explanation. Please provide these details. Also, the bottom phase images are not necessary and waste space. It would be far better to show higher power views of the data which is hard to see. Especially the 3 green cells at the 42-hour time point. The “data not shown” in line 230 need to be shown. This would address the point raised above.

Response of the authors:

The pictures are taken from the time-lapse movie (Supplementary film 1) made with a 20x objective to avoid too strong light exposure. One of the problems with this assay is that organoids grow in all directions, and very often they go out of focus. The fact that they do not appear to be the same optical section as the phase contrast images is due to the fact that the fluorescent images are at maximum fluorescence intensity (so the strongest signal per ROI from all 20 or such sections is selected and overlaid with all other sections) while the phase picture is only from the middle section. This is a technical issue which can not be resolved. We have provided higher resolution images in Figure 2 and we added a magnification in Figure S3a in the revised manuscript. Figure S3b shows the "data previously not shown".

5) A merged image of Figure 3A would help see the 3 cells we're meant to focus on. Also, there appear to be other cells that could have been labeled with arrows, that were not. Please also use arrows in the DAPI channel. On line 250 the "OSCAR^{high} (Ven90^{high} mCherry^{low}) cells the claim about cells being mCherry^{low} is subtle and very difficult to see, at least for this reviewer. One wonders if the images are overly saturated. Can the fluorescent data be presented in a complementary way to make the differences clearer visually? Also, the authors should consider including the raw data for each cell analyzed as supplemental information. The number of independent experiments and animals should also be included in the legend, not simply the number of cells analyzed.

Response of the authors:

We thank the reviewer for the helpful comments. We have added the merged image in the Figure 3a of the revised manuscript to help the visualization of the differences clearer. We also agree that there could be other cells that could have been labeled, as for example, the cell just below the two cells on the right. The three labeled with arrowheads are examples. We have also provided the raw data for each cell analyzed in Supplementary Table 1 of the revised manuscript and included the number of independent experiments and animals in the legend of the revised manuscript.

Additionally, we recognized an accidental mistake on Figure 3c that was showing the OSCAR signal normalized on the RNAPII-pSer2 signal (OSCAR/PolIII) instead of the correct OSCAR signal (Ven90/mCherry). We plotted the correct OSCAR signal (Ven90/mCherry) in the revised manuscript that continues to be significantly higher in RNAPII-pSer2^{low} cells than RNAPII-pSer2^{high} cells and does not change the finding. We apologize for the inconvenience.

6) The authors note that the small intestine of EF1a-OSCAR mice was fixed in 4% PFA for only 20 min at room temperature because the endogenous OSCAR fluorescence is very sensitive to fixation. Because the analysis is done using PFA fixed frozen sections, there should be a discussion of how tissue autofluorescence was handled. In addition, because all of the quantitative data presented in Figure 3b-g are dependent on the measurement of endogenous fluorescence of individual cells, a more detailed analysis of how these data were collected, and how the OSCAR ratio was calculated for each cell, is

essential for the reader to understand the strengths and limitations of the data. As presented it would be difficult for another group to repeat these studies.

Response of the authors:

We did negative controls from wt mouse intestine by using the same protocol as for the OSCAR mouse (see figure here below).

As expected the DAPI (blue channel) stains cell nuclei. Green oversaturated fluorescence shows some signal from cytoplasm and membranes of cells and also some nuclear stain in immune cells inside the villi, but was never observed in the nucleus of crypt epithelial cells of wt mice, unlike OSCAR mice. OSCAR nuclear signal was actually a quality mark for us during protocol implementation, especially fixation's time improvement. Similarly, red autofluorescence in wt mice was observed in granules in the paneth cell zone (as previously reported PMID: 29458079) and some signal from cell membrane, but never in the cytoplasm of crypt cells in wild type mouse. This was also mirrored by the exposure time. Usually, we would increase the exposure until some areas would be saturated. This was soon the case with endogenous red autofluorescence in the paneth cell zone in wt mouse. OSCAR mice had a saturation in other areas/cells, showing that the fluorescence signal stems from the OSCAR system, not from autofluorescence/paneth cell granules. Overall OSCAR signal was thus stronger than autofluorescence with the implemented protocol.

We also added detailed description of images quantification and OSCAR ratio calculation in the material and methods of the revised manuscript.

7) Figure 4a and b show the relative expression of Ven90 and mCherry using flow cytometry. I find it curious why mCherry would be expressed over a larger dynamic range (~2.5 logs) than Ven90 (~1.5 logs). Also, the diagonal nature of the data raise questions about whether some of the signal may be explained by autofluorescence.

Response of the authors:

We thank the reviewer for the comment. We have measured the fluorescence range of the two fluorescent proteins (see picture below). When considering the cells that have been sorted and further analyzed (the 5 gates that represent more >99% of cells), VeN90 has a range of 1.70 Log and mCherry of 2.21, confirming the observation of the reviewer that mCherry shows a larger dynamic range than VeN90. When considering only the cells of the main cell population) this difference is no longer visible (the inner square in the picture below). One hypothesis would be that VeN90 is more compensated by double (transcriptional and post-translational) regulation and the less is expressed (due to inactive RNAPII) the more it becomes brilliant (due to CDK9 inactivity); this could explain reduced fluorescence VeN90 range compared to mCherry.

We think that the diagonal nature is mainly the result of the fact that mCherry-P2A-VeN90 is transcribed as a single mRNA and is cleaved co-translationally, resulting in two proteins in a 1:1 stoichiometric ratio from one mRNA. We can also exclude autofluorescence, because we used a negative controls and set the gates to positive events based on the negative control and the two channels do not even need compensation (see the response to the comment 8 of the same reviewer).

8) The authors should show the complete set up of the FACS experiments including proper color controls (unstained, single colors, fluorescence minus one) and include the viability staining as well as FSC/SCC and doublet analysis, etc. Single color controls will require separate mice expressing mCherry or VeN90 alone, which are necessary to

properly set gates and allow for compensation, etc. Wildtype organoids should be used for the unstained, viability and EpCAM controls.

Response of the authors:

We agree with the reviewer and we have shown the complete gating strategy for isolating intestinal cells from OSCAR mice in the revised manuscript. The gating includes FSC/SSC, doublet cells removal, viability (DAPI) and Epcam+ (epithelial cells marker) staining. We added FACS profiles in Figure S4B). Unfortunately, we have not generated single-color (mCherry or VeN90) transgenic mice because we think we do not need compensation in the FACS experiments for the two fluorescent proteins mCherry and VeN90. We have tested several times the single fluorophores when we tested the OSCAR construct *in vitro* by transducing intestinal organoids and normal cells with single-color expressing vectors. Here below a FACS profile of a human cell line infected with the construct expressing either VeN90 (panel A) or mCherry (panel B). The analysis shows that the VeN90 signal measured in the GFP-A channel (panel A) does not lead to any signal in the PE-A channel and, vice-versa, the mCherry signal measured in the PE-A channel (panel B) does not lead to any signal in GFP-A channel.

9) On line 894 the figure legend states that “Population P1 and P3 contain OSCAR^{low} cells, while P2 contains OSCAR^{high} cells.” The fluorescence intensity of Ven90 for P2 and P3 are essentially the same. What distinguishes them better is their mCherry expression (high-medium-low).

Response of the authors:

We agree with the comment of the reviewer and we re-phrased the sentence to distinguish populations with the same OSCAR value in the revised manuscript.

10) Regarding the RNA-seq data presented in Figure 4 and 5, it is understandable why the authors would rely on 2 biological replicates. However, these data should be considered as screening data and not overly interpreted at the single gene level, as was done in 5b. Rather, conclusions about single genes should be performed and validated on independently sorted samples with larger replicate numbers using qRT-PCR analysis.

Response of the authors:

We fully agree with the reviewer and we performed qRT-PCR validation of the markers in the revised manuscript (Figure S4c).

11) Consistent with the prior commentxx , it is not OK to make statements such as this on line 305: "Remarkably, the P2 population did not show any enrichment of these known markers, except a slight enrichment of stem cell markers (Olfm4 and Ascl2) in one of the two replicates (Figure 5B, top-left panels)." This is the equivalent of showing an n=1 out of 2 samples analyzed.

Response of the authors:

We fully agree with the reviewer, we performed qRT-PCR validation of the markers in the revised manuscript (Figure S4c) and we re-phrased the sentence.

12) It would be interesting to see what would happen to the OSCAR populations in an injury model (e.g. radiation, when Lgr5 cells are depleted/damaged).

Response of the authors:

As previously written in the response for reviewer 1, we agree with the reviewer that resistance to high levels of DNA damage followed by cell cycle entry and regeneration represents an hallmark of the reserve stem cell population. To address this point, we performed total body gamma irradiation on OSCAR mice and measured the apoptosis marker Annexin V by FACS analysis. We found that P4/P5 cells (putative ISCs) have higher percentage of AnnexinV+ cells than P2 cells indicating that the OSCAR^{high} cells are more resistant to DNA damage following irradiation (Figure 6h of the revised manuscript).

13) As above, the gene expression data for Tert and Krt19 in the P2 population presented in Figure 6C should be performed and validated on independent samples.

Response of the authors:

We fully agree with the reviewer, we performed qRT-PCR validation of the markers in the revised manuscript (Figure S5a).

Minor comments:

Regarding the observation that TERT/GFP cells are rare, I believe that the original description of 1 cell per 150 crypts in REF 10 was based on tissue sectioning. From my reading of REF 11 it would appear that this conclusion has been updated using whole crypt analysis to reveal that the majority of crypts have 1 or 2 cells positive cells.

Response of the authors:

We agree with the reviewer that the conclusion has been updated in REF11. In introduction of REF11, the authors defined d-ISCs as "slowly cycling, relatively dormant ISCs (d-ISCs) marked by mTert (telomerase), Bmi1, Lrig1, HopX and Dclk1". They stated that "while there are between twelve and sixteen r-ISCs in a single small intestinal crypt, only one to two d-ISCs are typically present, underscoring their reserve role in intestinal maintenance". However, by using "high-resolution, three-dimensional imaging technique that allows for the analysis of endogenous GFP fluorescence in freshly isolated intestinal crypts, we were able to identify at least one mTert-GFP+ cell in up to two-thirds of all crypts (Figures 1a, S1a)", suggesting that mTERT-GFP is still unable to mark all the potential d-ISCs in the intestinal crypts. We re-phrased the sentence and updated the definition in the revised manuscript with "This cell population (detectable in many, but not all the intestinal crypts) is distinct....."

Lines 187-188 are awkward and hard to understand.

Response of the authors:

We re-phrased the sentence in the revised manuscript with "Increase of CDK9 activity leads to both increase in transcription and phosphorylation of the transgene VeN90; while phosphorylation of VeN90 leads to decreased fluorescence, that may be masked by a higher level of VeN90 transcripts (and relative proteins) due to an increased transcription. To properly work as a CDK9 reporter, the fluorescence of the protein needs to be normalized to expression"

Please provide split channel images for Figure 1A so the nuclear staining can be seen clearly.

Response of the authors:

We added split channels of Figure 1a in Figure S1b of the revised manuscript.

In Figure 1C please define what the filled arrowhead refers to.

Response of the authors:

We defined the filled arrowhead of figure 1c in the figure legend of the revised manuscript.

In Figure 3F/G, please provide representative pictures of Ki67 staining to accompany the figure.

Response of the authors:

As previously written in the point 2 of the major comments from the same reviewer, we have added representative image of Ki67 staining on intestinal crypt of OSCAR mice in Figure S4a.

On Line 161 remove reference to Figure S2A.

Response of the authors:

We removed reference to Figure S2a in the revised manuscript.

On Page 5 the authors define use in vivo when analyzing human melanoma cell lines. Please correct.

Response of the authors:

We removed the "in vivo" word in the revised manuscript.

The reference in line 80 needs to be fixed.

Response of the authors:

We fixed the reference in the revised manuscript.

Reviewer #3 (Remarks to the Author):

This manuscript describes OSCAR, an innovative approach to identifying cells with low transcriptional activity. The authors have also created a mouse expressing OSCAR in all cells. The cells demonstrated to have high OSCAR/low transcriptional activity do not belong to identifiable differentiated cells of the intestine and the authors propose that these cells are dormant stem cells. There remain several questions that the authors should address in a revised manuscript.

Response of the authors:

We thank the reviewer for the questions raised about our manuscript. Here below, a point-by-point response to his/her questions.

1) During the cell cycle, transcription decreases drastically during mitosis. Are the cells that were identified by the authors as dormant actually undergoing mitosis, with this stage of the cell cycle being the reason for lower transcriptional activity rather than the proposed dormancy? Can the authors distinguish low transcriptionally active mitotic cells from dormant cells?

Response of the authors:

To clarify this aspect, we performed Hoechst FACS cell cycle analysis and showed the results in the revised manuscript (Figure 5f,g). The experiment indicated that the P2 population contains mostly cells in G1 phase of the cell cycle (98.9±0.7% of cells in G1 phase, 1.02±0.7% of cells in S phase, 0.05±0.1% of cells in G2 phase), while cells in G2 phase of the cell cycle are most abundant in the P5 population. Therefore, yes, we can distinguish mitotic cells from dormant cells.

2) LGR5 is reported by the authors to marks highly cycling stem cells. Data for the expression level of this gene should be reported in figs. 5 and 6 along with genes that identify other intestinal crypt cell types.

Response of the authors:

We fully agree with the reviewer, and to address this concern we performed qRT-PCR validation of the markers and included Lgr5 in the revised manuscript (Figure S4c).

Minor comments

In a revised version of this manuscript, the authors should correct the reference citation on line 396 and make more consistent the reference throughout the manuscript to the OSCAR fluorescence ratio, which is variably referred to as OSCAR, OSCAR ratio fluorescence and OSCAR fluorescence throughout the manuscript.

Response of the authors:

We fixed the reference citation and the definition of OSCAR in the revised manuscript. We defined "OSCAR fluorescence ratio (defined as VeN90/mCherry fluorescence - abbreviated as OSCAR)", and kept this terminology along the whole manuscript.

On line 877, the statement "while the 10% of OSCAR^{low} cells..." is unclear and should be corrected.

Response of the authors:

We corrected the legend in the revised manuscript in " while the top 10% of OSCAR^{low} cells".

Reviewers' Comments:

Reviewer #1:

Remarks to the Author:

Review of OSCAR Nature Communications revision.

I address the points emanating from the original review and the author's response:

In regard to my initial comment #1 on the use of 'dormancy' and 'quiescence' to define distinct populations of cells in vivo. The authors response to this is convoluted and the text remains confusing. I simply don't understand why the author feel a need to present the concepts this way, as it has little bearing on the study at hand. In the field of somatic stem cell biology, which is what this study is addressing, quiescence=dormant. Satellite cells in the muscle, long-term HSCs, etc. The repeated use of the phrase 'quiescent but activated' makes absolutely no sense based on the textbook definition of these terms. This 'quiescent but activated' description of P3 is bizarre- how are these cells quiescent? They have lots of RNA, lots of markers of functional, differentiated cells- what about them is quiescent? These simply look like a mixed population of differentiated secretory cells. Again the authors seem to use quiescent to refer to cells that are transcriptionally active but not cycling. Based on that definition, 99.99% of the cells in our bodies are quiescent, and that's clearly not an accepted notion.

Ultimately, this is simply semantics, call them dormant, call them quiescent, call them G0, I don't care, just define what you mean by the term in the paper. Why the revised paper continues to compare and contrast 'dormancy' and 'quiescence' is beyond me. What reason is there for this? It simply creates confusion. There does not exist any quantitative, broadly accepted distinction between these terms. The cells in P2 appear dormant, quiescent, in G0, whatever. But to call P2 'dormant' and P2 'quiescent' is completely confusing.

This doesn't need to be complicated- the authors create a nice tool for isolating cells based on their global levels of transcription. That's it, nothing more, nothing less. The experiments in Figure 4 where arbitrary gates are drawn around cells in scatterplots, then assigned to groups based on bulk expression of some marker genes is a bit forced and the conclusions are not supported by the data- e.g., "these analyses indicated that P1, P3, P4 and P5 cell populations contain intestinal cells of an exclusive cell lineage or cell-cycle stage"- there is no evidence for this given the bulk transcriptome analysis in the absence of scRNAseq to clearly determine how specific these arbitrarily drawn gates are for any population. This type of language goes on in the rebuttal: "We agree with the reviewer that our tool is able to distinguish between transcriptionally inactive and active cells, with 1) a subgroup of reserve ISCs being transcriptionally inactive (OSCAR high - P2 population), supposedly in G0, and 2) another subgroup of reserve ISCs (OSCAR low - P3 population), transcriptionally active and characterized by secretory lineage markers." Without scRNAseq this is largely overinterpreted. scRNAseq would enable the authors to ascribe identity and heterogeneity to these populations, otherwise they are best of referring to an 'enrichment' of cells in these various lineages/states. It's highly unlikely these are homogenous populations.

Most importantly, the authors repeatedly refer to various populations as 'reserve stem cells', but provide no evidence for this. A 'reserve stem cell' is generally defined by its ability to regenerate the epithelium in response to killing of active stem/progenitor/TA cells by injury (usually DNA damage in the literature). No such assays are performed here, nor can they be in the absence of a lineage tracing tool, and thus statements like "In particular, this reporter distinguished between dormant "reserve" stem-like cells and quiescent, but activated secretory lineage precursor cells in the small intestine" cannot be justified".

Overall, the tool is nice, the addition of the RNA content and cell death assays are nice. But ultimately the written document is excessively confusing and data are overinterpreted.

A specific question: Figure 5h, the new data on RNA content- I am assuming this was measured using the same number of cells in each group? This should be explicitly stated (I didn't see it methods or legend, results just says 'normalized to the number of cells', and I'm not clear on what that means.

Reviewer #2:

Remarks to the Author:

The authors have address my primary concerns in a robust way. I especially appreciate the radiation experiment presented in Fig 6. My only concern about these data, though, is the placement of the gates in Fig 6g, given the larger number of cells in the p4/5 population, which could skew the overall percentages of + cells. Thus, slightly more conservative gating (for all three plots) would likely lead to a similar, but more convincing result.

Reviewer #3:

Remarks to the Author:

The authors have fully responded to the reviewers' comments and clarified a number of points with the result that I have no other concerns about the manuscript and its content. I believe that this interesting work will be of interest to a number of readers of this journal and recommend that it be accepted for publication.

Dear Editor,
please find here below a point-by-point response to the reviewer's comments.

REVIEWER COMMENTS

Reviewer #1 (Remarks to the Author):

Review of OSCAR Nature Communications revision.

I address the points emanating from the original review and the author's response:

In regard to my initial comment #1 on the use of 'dormancy' and 'quiescence' to define distinct populations of cells in vivo. The authors response to this is convoluted and the text remains confusing. I simply don't understand why the author feel a need to present the concepts this way, as it has little bearing on the study at hand. In the field of somatic stem cell biology, which is what this study is addressing, quiescence=dormant. Satellite cells in the muscle, long-term HSCs, etc. The repeated use of the phrase 'quiescent but activated' makes absolutely no sense based on the textbook definition of these terms. This 'quiescent but activated' description of P3 is bizarre- how are these cells quiescent? They have lots of RNA, lots of markers of functional, differentiated cells- what about them is quiescent? These simply look like a mixed population of differentiated secretory cells. Again the authors seem to use quiescent to refer to cells that are transcriptionally active but not cycling. Based on that definition, 99.99% of the cells in our bodies are quiescent, and that's clearly not an accepted notion.

Ultimately, this is simply semantics, call them dormant, call them quiescent, call them G0, I don't care, just define what you mean by the term in the paper. Why the revised paper continues to compare and contrast 'dormancy' and 'quiescence' is beyond me. What reason is there for this? It simply creates confusion. There does not exist any quantitative, broadly accepted distinction between these terms. The cells in P2 appear dormant, quiescent, in G0, whatever. But to call P2 'dormant' and P2 'quiescent' is completely confusing.

Response of the authors:

We further adjusted the introduction, results and discussion section to completely remove the comparison of dormancy and quiescence as requested by the reviewer.

This doesn't need to be complicated- the authors create a nice tool for isolating cells based on their global levels of transcription. That's it, nothing more, nothing less. The experiments in Figure 4 where arbitrary gates are drawn around cells in scatterplots, then assigned to groups based on bulk expression of some marker genes is a bit forced and the conclusions are not supported by the data- e.g., "these analyses indicated that P1, P3, P4 and P5 cell populations contain intestinal cells of an exclusive cell lineage or cell-cycle stage"- there is no evidence for this given the bulk transcriptome analysis in the absence of scRNAseq to clearly determine how specific these arbitrarily drawn gates are for any population.

Response of the authors:

We further modified the results section and changed " contain intestinal cells of an exclusive cell lineage or cell-cycle stage" with "are enriched with intestinal cells of specific cell lineage or cell-cycle stage" as suggested by the reviewer.

Most This type of language goes on in the rebuttal: "We agree with the reviewer that our tool is able to distinguish between transcriptionally inactive and active cells, with 1) a subgroup of reserve ISCs being transcriptionally inactive (OSCAR high - P2 population), supposedly in G0, and 2) another subgroup of reserve ISCs (OSCAR low - P3 population), transcriptionally active and characterized by secretory lineage markers." Without scRNAseq this is largely overinterpreted. scRNAseq would enable the authors to ascribe identity and heterogeneity to these populations, otherwise they are best of referring to an 'enrichment' of cells in these various lineages/states. It's highly unlikely these are homogenous populations. importantly, the authors repeatedly refer to various populations as 'reserve stem cells', but provide no evidence for this. A 'reserve stem cell' is generally defined by its ability to regenerate the epithelium in response to killing of active stem/progenitor/TA cells by injury (usually DNA damage in the literature). No such assays are performed here, nor can they be in the absence of a lineage tracing tool, and thus statements like "In particular, this reporter distinguished between dormant "reserve" stem-like cells and quiescent, but activated secretory lineage precursor cells in the small intestine" cannot be justified".

Response of the authors:

We further corrected the results and discussion section and removed the comparison and any statements between P2 and P3 population.

Overall, the tool is nice, the addition of the RNA content and cell death assays are nice. But ultimately the written document is excessively confusing and data are overinterpreted.

A specific question: Figure 5h, the new data on RNA content- I am assuming this was measured using the same number of cells in each group? This should be explicitly stated (I didn't see it methods or legend, results just says 'normalized to the number of cells', and I'm not clear on what that means.

We thank the reviewer for the question. The total amount of RNA measured by Qubit from each group was divided by the number of cells of each group that could be different depending on the sorting. Then all the values were normalized to the group replicate 1 of the P1 population to show a relative fold change. We further modified the methods section and better explained how the analysis was performed.

Reviewer #2 (Remarks to the Author):

The authors have address my primary concerns in a robust way. I especially appreciate the radiation experiment presented in Fig 6. My only concern about these data, though, is the placement of the gates in Fig 6g, given the larger number of cells in the p4/5 population, which could skew the overall percentages of + cells. Thus, slightly more conservative gating (for all three plots) would likely lead to a similar, but more convincing result.

Response of the authors:

We thank the reviewer for the comment. We employed a more conservative gating in Figure 6 of the revised manuscript as suggested by the reviewer. The new calculation led to a similar ratio between the P2 and the P4/5 populations and does not affect the result.

Reviewer #3 (Remarks to the Author):

The authors have fully responded to the reviewers' comments and clarified a number of points with the result that I have no other concerns about the manuscript and its content. I believe that this interesting work will be of interest to a number of readers of this journal and recommend that it be accepted for publication.

Response of the authors:

We thank the reviewer for the comment.

Reviewers' Comments:

Reviewer #1:

Remarks to the Author:

Thank you for cleaning up the language in this last revision. I think it now presents a much more clear and less biased presentation of this novel tool.